# Nonconvex Stochastic Optimization under Heavy-Tailed Noises: Optimal Convergence without Gradient Clipping

**Zijian Liu**[*] **& Zhengyuan Zhou**
Stern School of Business, New York University
`{zl3067,zzhou}@stern.nyu.edu`

## Abstract

Recently, the study of heavy-tailed noises in first-order nonconvex stochastic optimization has gotten a lot of attention since it was recognized as a more realistic condition as suggested by many empirical observations. Specifically, the stochastic noise (the difference between the stochastic and true gradient) is considered to have only a finite $\mathfrak{p}$-th moment where $\mathfrak{p} \in (1, 2]$ instead of assuming it always satisfies the classical finite variance assumption. To deal with this more challenging setting, people have proposed different algorithms and proved them to converge at an optimal $\mathcal{O}(T^{\frac{1-\mathfrak{p}}{3\mathfrak{p}-2}})$ rate for smooth objectives after $T$ iterations. Notably, all these new-designed algorithms are based on the same technique – gradient clipping. Naturally, one may want to know whether the clipping method is a necessary ingredient and the only way to guarantee convergence under heavy-tailed noises. In this work, by revisiting the existing Batched Normalized Stochastic Gradient Descent with Momentum (Batched NSGDM) algorithm, we provide the first convergence result under heavy-tailed noises but *without* gradient clipping. Concretely, we prove that Batched NSGDM can achieve the optimal $\mathcal{O}(T^{\frac{1-\mathfrak{p}}{3\mathfrak{p}-2}})$ rate even under the relaxed smooth condition. More interestingly, we also establish the first $\mathcal{O}(T^{\frac{1-\mathfrak{p}}{2\mathfrak{p}}})$ convergence rate in the case where the tail index $\mathfrak{p}$ is unknown in advance, which is arguably the common scenario in practice.

## 1 Introduction

This paper studies the optimization problem $\min_{\boldsymbol{x} \in \mathbb{R}^d} F(\boldsymbol{x})$ where $F : \mathbb{R}^d \to \mathbb{R}$ is differentiable and could be nonconvex. When $F$ is smooth (i.e., the gradient of $F$ is Lipschitz), the classical first-order method, Gradient Descent (GD), is known to converge at the optimal rate $\mathcal{O}(T^{-\frac{1}{2}})$ to find a stationary point (i.e., to minimize $\|\nabla F(\boldsymbol{x})\|$) (Nesterov et al., 2018). However, a main drawback of GD in the modern view is that it requires true gradients, which could be computationally burdensome (e.g., large-scale tasks) or even infeasible to obtain (e.g., streaming data). As such, a famous variant of GD, Stochastic Gradient Descent (SGD) (Robbins & Monro, 1951), has become the gold standard and been widely implemented nowadays due to its lightweight yet efficient computational procedure. Under the standard finite variance noise condition, i.e., the second moment of the difference between the stochastic gradient and the true gradient is bounded, SGD has been proved to converge in the rate of $\mathcal{O}(T^{-\frac{1}{4}})$ (Ghadimi & Lan, 2013), which is unimprovable if without further assumptions as indicated by the lower bound (Arjevani et al., 2023).

Although the finite variance assumption has been widely adopted in theoretical study (see, e.g., Lan (2020)), it has been recently recognized as too optimistic in modern machine learning tasks pointed out by empirical observations (Simsekli et al., 2019; Şimşekli et al., 2019; Zhang et al., 2020c), which reveal a more realistic setting: the heavy-tailed regime, i.e., the stochastic noise only has a finite $\mathfrak{p}$-th moment where $\mathfrak{p} \in (1, 2]$. Such a new assumption brings new challenges in both algorithmic design and theoretical analysis since SGD unfortunately fails to work and the prior theory of SGD becomes invalid when $\mathfrak{p} < 2$ (Zhang et al., 2020c).

---

[*]Corresponding author.

To resolve the failure of SGD under heavy-tailed noises, Clipped SGD (or its further variants) has been proposed and shown to converge in both expectation (Zhang et al., 2020c) and high probability (Cutkosky & Mehta, 2021; Liu et al., 2023b; Nguyen et al., 2023; Liu et al., 2024) at a rate $\mathcal{O}(T^{\frac{1-\mathfrak{p}}{3\mathfrak{p}-2}})$, which is indeed optimal (Zhang et al., 2020c). As suggested by the name, the central tool lying in all these existing algorithms is gradient clipping, which seems to be a necessary ingredient against heavy-tailed noises so far. Therefore, we are naturally led to the following question:

*Is gradient clipping the only way to guarantee convergence under heavy-tailed noises? If not, can we find an algorithm that converges at the optimal rate $\mathcal{O}(T^{\frac{1-\mathfrak{p}}{3\mathfrak{p}-2}})$ without clipping?*

Another important but often omitted issue in previous studies is that the tail index $\mathfrak{p}$ of heavy-tailed noises is always implicitly assumed to be known and then used to choose the clipping magnitude and the stepsize (or the momentum parameter) (Zhang et al., 2020c; Cutkosky & Mehta, 2021; Liu et al., 2023b; Nguyen et al., 2023; Liu et al., 2024). However, knowing $\mathfrak{p}$ exactly or even estimating its approximate value is a non-trivial task in lots of cases, e.g., the online setting. Consequently, the existing convergence theory for Clipped SGD immediately becomes vacuous when no prior information on $\mathfrak{p}$ is guaranteed, which is however arguably the common scenario in practice. The above discussion thereby leads us to another research question:

*Does there exist an algorithm that provably converges under heavy-tailed noises even if the tail index $\mathfrak{p}$ is unknown?*

This work provides affirmative answers to both of the above questions by revisiting a well-known technique in optimization: gradient normalization, which is surprisingly effective even under heavy-tailed noises as demonstrated by our refined theoretical analysis.

## 1.1 Our Contributions

We study the Batched Normalized Stochastic Gradient Descent with Momentum (Batched NSGDM) algorithm (Cutkosky & Mehta, 2020) under heavy-tailed noises and establish several new results:

- We prove that Batched NSGDM converges in expectation at the optimal rate $\mathcal{O}(T^{\frac{1-\mathfrak{p}}{3\mathfrak{p}-2}})$ when noises only have a finite $\mathfrak{p}$-th moment where $\mathfrak{p} \in (1, 2]$, which is the first convergence result under heavy-tailed noises not requiring gradient clipping[1].

- We establish a refined lower complexity result having a more precise order on problem-dependent parameters (e.g., the noise level $\sigma_0$), which perfectly matches our new convergence theory for Batched NSGDM further showing the optimality of our analysis.

- We initiate the study of optimization under heavy-tailed noises with an unknown tail index $\mathfrak{p}$ and provide the first provable rate $\mathcal{O}(T^{\frac{1-\mathfrak{p}}{2\mathfrak{p}}})$ also achieved by the same Batched NSGDM method indicating the robustness of gradient normalization against heavy-tailed noises[2].

- Our analysis goes beyond the classical smoothness condition and heavy-tailed noises assumption studied in previous works and is the first to hold under their generalized counterparts (see Assumptions 2.2 and 2.4).

- Our proof is based on a novel expected inequality for the vector-valued martingale difference sequence, which might be of independent interest.

## 1.2 Related Work

We focus on the literature on nonconvex problems under heavy-tailed noises. For recent progress on convex optimization, the reader can refer to Zhang & Cutkosky (2022); Sadiev et al. (2023); Liu & Zhou (2023); Kornilov et al. (2023); Puchkin et al. (2024); Gorbunov et al. (2024) for details.

**Upper bound under heavy-tailed noises.** For smooth objectives, different works have established the optimal rate $\mathcal{O}(T^{\frac{1-\mathfrak{p}}{3\mathfrak{p}-2}})$ (up to extra logarithmic factors) for Clipped SGD or its further variants

---

[1]In preparing the camera-ready version of the paper, it was brought to our attention that two independent and concurrent works established similar results (Hübler et al., 2024; Sun et al., 2024).

[2]Hübler et al. (2024) also gave a matched lower bound for Batched NSGD, i.e., $\beta_t \equiv 0$ in Algorithm 1.

(Zhang et al., 2020c; Cutkosky & Mehta, 2021; Liu et al., 2023b; Nguyen et al., 2023; Liu et al., 2024), among which, Zhang et al. (2020c) and Nguyen et al. (2023) respectively provide the best expected and high-probability bounds for Clipped SGD as their results do not contain any extra $\mathcal{O}(\log T)$ factor. Notably, Zhang et al. (2020c) can recover the standard $\mathcal{O}(T^{-\frac{1}{2}})$ rate in the noiseless case. In contrast, the rates by Cutkosky & Mehta (2021); Liu et al. (2023b); Nguyen et al. (2023); Liu et al. (2024) are not adaptive to the noise level. In addition, we note that the results from Cutkosky & Mehta (2021) also depend on an extra vulnerable assumption, i.e., the stochastic gradient itself has a finite $\mathfrak{p}$-th moment, which can be easily violated. However, all these works require a known tail index $\mathfrak{p}$ to establish convergence theory, which may not be realistic in practice.

**Lower bound under heavy-tailed noises.** As far as we know, Zhang et al. (2020c) is the first and only work that provides the $\Omega(T^{\frac{1-\mathfrak{p}}{3\mathfrak{p}-2}})$ lower bound for nonconvex optimization under the classical smooth assumption and the finite $\mathfrak{p}$-th moment condition on noises, where the proof is based on the tool named probability zero-chain developed by Arjevani et al. (2023). However, we should remind the reader that the lower bound by Zhang et al. (2020c) is actually proved based on assuming a finite $\mathfrak{p}$-th moment on the stochastic gradient instead of the noise and hence may fail to provide a correct dependence on problem-dependent parameters like the noise level $\sigma_0$.

In addition, we provide a quick review of the gradient normalization technique.

**Gradient normalization.** The normalized gradient method has a long history and could date back to the pioneering work of Nesterov (1984), which is the first paper to suggest considering normalization in (quasi-)convex optimization problems and provides a theoretical convergence rate. Many later works (e.g., Kiwiel (2001); Hazan et al. (2015); Levy (2016); Nacson et al. (2019)) further explore the potential of gradient normalization. In deep learning, gradient normalization (or its variant) has also gained more and more attention since it can tackle the gradient explosion/vanish issue and has been observed to accelerate the training (You et al., 2017; 2019). However, a provable theory still lacks for general nonconvex problems until (Cutkosky & Mehta, 2020), who established the first meaningful bound by adding momentum to the normalized gradient method.

## 2 PRELIMINARIES

**Notation.** $\mathbb{N}$ denotes the set of natural numbers (excluding 0). $[T] \triangleq \{1, 2, \cdots, T\}$ for any $T \in \mathbb{N}$. $\langle \cdot, \cdot \rangle$ is the Euclidean inner product on $\mathbb{R}^d$ and $\|\cdot\| \triangleq \sqrt{\langle \cdot, \cdot \rangle}$ is the $\ell_2$ norm. Given a sequence $r_t \in \mathbb{R}, \forall t \in [T]$, we use the notation $r_{s:t} \triangleq \prod_{\ell=s}^{t} r_\ell$ for any $1 \leq s \leq t \leq T$ and $r_{s:t} \triangleq 1$ if $s > t$.

We consider the optimization problem $\min_{\boldsymbol{x} \in \mathbb{R}^d} F(\boldsymbol{x})$ in this work, where $F : \mathbb{R}^d \to \mathbb{R}$ is differentiable and potentially nonconvex. A computationally tractable goal is to minimize $\|\nabla F(\boldsymbol{x})\|$, which is the focus hereinafter. Our analysis relies on the following assumptions.

**Assumption 2.1.** *Finite Lower Bound.* $F_* \triangleq \inf_{\boldsymbol{x} \in \mathbb{R}^d} F(\boldsymbol{x}) > -\infty$.

**Assumption 2.2.** *Generalized Smoothness.* *There exist $L_0 \geq 0$ and $L_1 \geq 0$ such that $\|\nabla F(\boldsymbol{x}) - \nabla F(\boldsymbol{y})\| \leq (L_0 + L_1 \|\nabla F(\boldsymbol{x})\|) \|\boldsymbol{x} - \boldsymbol{y}\|$ for any $\boldsymbol{x}, \boldsymbol{y} \in \mathbb{R}^d$ satisfying $\|\boldsymbol{x} - \boldsymbol{y}\| \leq \frac{1}{L_1}$.*

The original definition of generalized smoothness was proposed by Zhang et al. (2020b) but required the objective to be twice differentiable. Here, we instead adopt a weaker version introduced later by Zhang et al. (2020a), which only needs $F$ to be differentiable. As one can see, Assumption 2.2 degenerates to the standard $L_0$-smoothness when $L_1 = 0$ and hence is more general. We note that there exist other versions of generalized smoothness proposed recently and refer to Chen et al. (2023); Li et al. (2023a;b) for details.

**Assumption 2.3.** *Unbiased Estimator.* *At the $t$-th iteration, we can access a batch of unbiased gradient estimator $G_t \triangleq \{\boldsymbol{g}_t^1, \cdots, \boldsymbol{g}_t^B\}$, i.e., $\mathbb{E}[\boldsymbol{g}_t^i \mid \mathcal{F}_{t-1}] = \nabla F(\boldsymbol{x}_t), \forall i \in [B]$, where $B$ is the batch size and $\mathcal{F}_t \triangleq \sigma(G_1, \cdots, G_t)$ denotes the natural filtration. Moreover, we assume that $\boldsymbol{g}_t^i, \forall i \in [B]$ are mutually independent for any fixed $t$.*

**Assumption 2.4.** *Generalized Heavy-Tailed Noises.* *There exist $\mathfrak{p} \in (1, 2]$, $\sigma_0 \geq 0$ and $\sigma_1 \geq 0$ such that $\mathbb{E}\left[\|\boldsymbol{\xi}_t^i\|^{\mathfrak{p}} \mid \mathcal{F}_{t-1}\right] \leq \sigma_0^{\mathfrak{p}} + \sigma_1^{\mathfrak{p}} \|\nabla F(\boldsymbol{x}_t)\|^{\mathfrak{p}}, \forall i \in [B]$ almost surely where $\boldsymbol{\xi}_t^i \triangleq \boldsymbol{g}_t^i - \nabla F(\boldsymbol{x}_t)$.*

We remark that Assumption 2.4 is a relaxation of the traditional heavy-tailed noises assumption (i.e., set $\sigma_1 = 0$) and is new as far as we know. The reason for proposing this generalized version

is that the finite $\mathfrak{p}$-th moment requirement used in prior works can be violated in certain situations where the new assumption instead holds. We refer the reader to Example A.1 provided in Appendix A for details. In addition, such a form of Assumption 2.4 may reminisce about the affine variance condition studied in the existing literature (Bottou et al., 2018). Indeed, this new assumption is inspired by it and can also be viewed as its extension.

However, to make our work more reader-friendly, we will stick to the case $\sigma_1 = 0$ (i.e., the classical heavy-tailed noises assumption used in prior works) in the main text to focus on conveying our high-level idea and avoid further complicating the analysis. The full version of our new result considering an arbitrary pair $(\sigma_0, \sigma_1)$ in Assumption 2.4 is deferred to Appendix D.

To finish this section, we introduce the following smoothness inequality under Assumption 2.2, whose proof is omitted and can be found in, for example, Zhang et al. (2020a).

**Lemma 2.5.** *Under Assumption 2.2, for any $\boldsymbol{x}, \boldsymbol{y} \in \mathbb{R}^d$ satisfying $\|\boldsymbol{x} - \boldsymbol{y}\| \leq \frac{1}{L_1}$, there is*

$$F(\boldsymbol{y}) \leq F(\boldsymbol{x}) + \langle \nabla F(\boldsymbol{x}), \boldsymbol{y} - \boldsymbol{x} \rangle + \frac{L_0 + L_1 \|\nabla F(\boldsymbol{x})\|}{2} \|\boldsymbol{x} - \boldsymbol{y}\|^2 .$$

## 3 CONVERGENCE WITHOUT GRADIENT CLIPPING

---
**Algorithm 1** Batched Normalized Stochastic Gradient Descent with Momentum (Batched NSGDM)

---
**Input:** initial point $\boldsymbol{x}_1 \in \mathbb{R}^d$, batch size $B \in \mathbb{N}$, momentum parameter $\beta_t \in [0, 1]$, stepsize $\eta_t > 0$
**for** $t = 1$ **to** $T$ **do**
    $\boldsymbol{g}_t = \frac{1}{B} \sum_{i=1}^{B} \boldsymbol{g}_t^i$
    $\boldsymbol{m}_t = \beta_t \boldsymbol{m}_{t-1} + (1 - \beta_t)\boldsymbol{g}_t$          $\triangleright$ where $\boldsymbol{m}_0 \triangleq \boldsymbol{g}_1$
    $\boldsymbol{x}_{t+1} = \boldsymbol{x}_t - \eta_t \frac{\boldsymbol{m}_t}{\|\boldsymbol{m}_t\|}$          $\triangleright$ where $\frac{\boldsymbol{0}}{\|\boldsymbol{0}\|} \triangleq \boldsymbol{0}$
**end for**

---

*Remark* 3.1. Instead of the norm normalization in Algorithm 1, we can consider an elementwise normalization rule, i.e., $\boldsymbol{x}_{t+1}[i] = \boldsymbol{x}_t[i] - \eta_{t,i} \frac{\boldsymbol{m}_t[i]}{|\boldsymbol{m}_t[i]|}, \forall i \in [d]$, which is known as Batched Sign Stochastic Gradient Descent with Momentum (Batched SSGDM). By applying the idea introduced in Section 4, one can also establish the convergence of Batched SSGDM under heavy-tailed noises.

The method we are interested in, Batched NSGDM (Cutkosky & Mehta, 2020), is provided above in Algorithm 1. Compared to the widely used Batched SGDM algorithm, the only difference is the extra normalization step, which we will show has a crucial effect when dealing with heavy-tailed noises. Since many prior works (e.g., You et al. (2017); Cutkosky & Mehta (2020); Jin et al. (2021)) have explained how and why Batched NSGDM works (in the finite/affine variance case), we hence do not repeat the discussion here again. The reader seeking the intuition behind the algorithm could refer to, for example, Cutkosky & Mehta (2020) for details.

Now we are ready to present our new result for this classical algorithm under heavy-tailed noises.

### 3.1 CONVERGENCE WITH A KNOWN TAIL INDEX $\mathfrak{p}$

In this subsection, we provide the convergence rate of Batched NSGDM under an ideal situation, i.e., when every problem-dependent parameter is known, which is commonly assumed implicitly in the optimization literature.

**Theorem 3.2.** *Under Assumptions 2.1, 2.2, 2.3 and 2.4 (with $\sigma_1 = 0$), let $\Delta_1 \triangleq F(\boldsymbol{x}_1) - F_*$, then for any $T \in \mathbb{N}$, by taking*

$$\beta_t \equiv \beta = 1 - \min\left\{1, \max\left\{\left(\frac{\Delta_1 L_1 + \sigma_0}{\sigma_0 T}\right)^{\frac{\mathfrak{p}}{2\mathfrak{p}-1}}, \left(\frac{\Delta_1 L_0}{\sigma_0^2 T}\right)^{\frac{\mathfrak{p}}{3\mathfrak{p}-2}}\right\}\right\},$$

$$\eta_t \equiv \eta = \min\left\{\sqrt{\frac{(1-\beta)\Delta_1}{L_0 T}}, \frac{1-\beta}{8L_1}\right\}, \quad B = 1,$$

*Algorithm 1 guarantees*

$$\frac{1}{T}\sum_{t=1}^{T}\mathbb{E}\left[\|\nabla F(\boldsymbol{x}_t)\|\right] = \mathcal{O}\left(\frac{\Delta_1 L_1}{T} + \sqrt{\frac{\Delta_1 L_0}{T}} + \frac{(\Delta_1 L_1)^{\frac{\mathfrak{p}-1}{2\mathfrak{p}-1}}\sigma_0^{\frac{\mathfrak{p}}{2\mathfrak{p}-1}} + \sigma_0}{T^{\frac{\mathfrak{p}-1}{2\mathfrak{p}-1}}} + \frac{(\Delta_1 L_0)^{\frac{\mathfrak{p}-1}{3\mathfrak{p}-2}}\sigma_0^{\frac{\mathfrak{p}}{3\mathfrak{p}-2}}}{T^{\frac{\mathfrak{p}-1}{3\mathfrak{p}-2}}}\right).$$

Theorem 3.2 states the convergence rate of Algorithm 1 under heavy-tailed noises. The full version, Theorem D.1, that works for any $\sigma_1 \geq 0$ is deferred to Appendix D. As a quick sanity check, Theorem 3.2 reduces to the rate $\mathcal{O}((\Delta_1 L_0 \sigma_0^2/T)^{\frac{1}{4}})$ when $\mathfrak{p} = 2$ (i.e., the finite variance case), which is known to be tight not only in $T$ but also in $\Delta_1, L_0$ and $\sigma_0$ (Arjevani et al., 2023).

We would like to discuss this theorem here further. First and most importantly, as far as we know, this is the first and only convergence result for nonconvex stochastic optimization under heavy-tailed noises but without employing the gradient clipping technique, which is the central tool for all previous algorithms under the same setting, not to mention the rate $\mathcal{O}(T^{\frac{1-\mathfrak{p}}{3\mathfrak{p}-2}})$ is also optimal in $T$ since it matches the lower bound $\Omega(T^{\frac{1-\mathfrak{p}}{3\mathfrak{p}-2}})$ proved in Zhang et al. (2020c). In fact, the lower-order term $\mathcal{O}((\Delta_1 L_0 \sigma_0^{\frac{\mathfrak{p}}{\mathfrak{p}-1}}/T)^{\frac{\mathfrak{p}-1}{3\mathfrak{p}-2}})$ in our rate is also tight in $\Delta_1, L_0$ and $\sigma_0$ as indicated by Theorem 3.3 below, where we present a refined lower bound by extending the prior result (Zhang et al., 2020c). To the best of our knowledge, we are also the first to obtain a tight dependence on these parameters compared to the previous best-known bounds for Clipped SGD that are only optimal in $T$ (Zhang et al., 2020c; Nguyen et al., 2023). The proof of Theorem 3.3 is given in Appendix D and mostly follows the same way established in Carmon et al. (2020); Arjevani et al. (2023); Zhang et al. (2020c) but with a simple alteration to ensure that the noise magnitude $\sigma_0$ shows up in the correct order.

**Theorem 3.3.** *For any given $\mathfrak{p} \in (1, 2]$, $\Delta_1, L_0, \sigma_0 > 0$, and small enough $\varepsilon > 0$, there exist a function $F$ (depending on the previous parameters) satisfying Assumptions 2.1, 2.2 (with $L_1 = 0$) and $F(\mathbf{0}) - F_* \leq \Delta_1$ and a stochastic oracle satisfying Assumptions 2.3 (with $B = 1$) and 2.4 (with $\sigma_1 = 0$) such that any zero-respecting algorithm[3] startinig from $\boldsymbol{x}_1 = \mathbf{0}$ requires $\Omega(\Delta_1 L_0 \sigma_0^{\frac{\mathfrak{p}}{\mathfrak{p}-1}}\varepsilon^{-\frac{3\mathfrak{p}-2}{\mathfrak{p}-1}})$ iterations to find an $\varepsilon$-stationary point, i.e., $\mathbb{E}\left[\|\nabla F(\boldsymbol{x})\|\right] \leq \varepsilon$.*

*Remark* 3.4. The careful reader may find that Theorem 3.3 is established under the classical smooth assumption instead of the relaxed $(L_0, L_1)$-smooth condition. However, noticing that the function class satisfying the traditional smooth condition is a subclass of the function set fulfilling generalized smoothness, this lower bound can thus be directly applied to the setting considered in Theorem 3.2.

*Remark* 3.5. We also note that, when $\mathfrak{p} = 2$, Theorem 3.3 perfectly matches the existing lower bound $\Omega(\Delta_1 L_0 \sigma_0^2 \varepsilon^{-4})$ in the finite variance case (Arjevani et al., 2023).

Moreover, we would like to mention that Theorem 3.2 holds under generalized smoothness (and generalized heavy-tailed noises), which greatly extends the implication of our result compared to the previous works (Zhang et al., 2020c; Nguyen et al., 2023) that can be only applied to the classical setting. In addition, the reader may want to ask why we need a batch size $B$ given it is set to 1. The reason for considering $B$ is that it plays an important role when $\sigma_1 > 0$. To be precise, $B$ could possibly be larger than 1 to guarantee the convergence when $\sigma_1 > 0$. For details about how the batch size $B$ works, please refer to Theorem D.1 and its proof. Lastly, we want to point out that Theorem 3.2 perfectly recovers the fastest rate in the noiseless case. In other words, when $\sigma_0 = 0$, the rate degenerates to $\mathcal{O}(\Delta_1 L_1/T + \sqrt{\Delta_1 L_0/T})$, which is the best result for deterministic $(L_0, L_1)$-smooth nonconvex optimization as far as we are aware (Liu et al., 2023a).

Given the above discussion, we believe that our work provides a new insight on how to deal with heavy-tailed noises, i.e., using normalization, in stochastic nonconvex optimization problems.

## 3.2 Convergence without A Known Tail Index $\mathfrak{p}$

As pointed out at the beginning of Subsection 3.1, the convergence result shown in Theorem 3.2 can only hold under full information because the choices of $\beta$ and $\eta$ heavily rely on problem-dependent parameters, which are, however, hard to estimate in practice. Especially, assuming prior information on $\mathfrak{p}$ is not realistic. As such, we will try to reduce the dependence on these parameters in this subsection.

---

[3]A first-order algorithm is called zero-respecting if it satisfies $\boldsymbol{x}_t \in \cup_{s<t}\text{support}(\boldsymbol{g}_s), \forall t \in \mathbb{N}$. The reader could refer to Definition 1 in Arjevani et al. (2023) for details.

**Theorem 3.6.** *Under Assumptions 2.1, 2.2, 2.3 and 2.4 (with $\sigma_1 = 0$), let $\Delta_1 \triangleq F(\boldsymbol{x}_1) - F_*$, then for any $T \in \mathbb{N}$, by taking*

$$\beta_t \equiv \beta = 1 - \frac{1}{T^{\frac{1}{2}}}, \quad \eta_t \equiv \eta = \min\left\{\frac{1}{T^{\frac{3}{4}}}, \frac{1}{8L_1 T^{\frac{1}{2}}}\right\}, \quad B = 1,$$

*Algorithm 1 guarantees*

$$\frac{1}{T}\sum_{t=1}^{T}\mathbb{E}\left[\|\nabla F(\boldsymbol{x}_t)\|\right] = \mathcal{O}\left(\frac{\Delta_1 L_1}{\sqrt{T}} + \frac{\Delta_1 + L_0}{T^{\frac{1}{4}}} + \frac{\sigma_0}{T^{\frac{\mathfrak{p}-1}{2\mathfrak{p}}}}\right).$$

Theorem 3.6 shows the first provable convergence upper bound $\mathcal{O}(T^{\frac{1-\mathfrak{p}}{2\mathfrak{p}}})$ when the tail index $\mathfrak{p}$ is unknown and, at the same time, tries to relax the dependence on other parameters. As one can see, the only parameter that we require now is $L_1$. It is noteworthy that, once the time horizon $T$ is large enough to satisfy $T = \Omega(L_1^4)$ (or equivalently, $L_1$ is small enough to satisfy $L_1 = \mathcal{O}(T^{\frac{1}{4}})$), we can get rid of $L_1$ in $\eta$ and thus do not need any information on the problem. A remarkable implication is that, in the same classical smooth case (i.e., when $L_1 = 0$) studied in previous works (Zhang et al., 2020c; Nguyen et al., 2023), Batched NSGDM can converge in the rate $\mathcal{O}(T^{\frac{1-\mathfrak{p}}{2\mathfrak{p}}})$ without knowing any of $\mathfrak{p}, \Delta_1, L_0, \sigma_0$. In contrast, the choices of the clipping magnitude and the stepsize in Clipped SGD provided by Zhang et al. (2020c); Nguyen et al. (2023) heavily depend on these parameters.

In addition, there are several points we would like to clarify. First, one may want to ask whether the requirement of knowing $L_1$ can be totally lifted, which we incline to a positive answer. But to prevent deviating from the main topic of our paper — heavy-tailed noises, we defer the detailed discussion about $L_1$ to Appendix B, in which we will explain the reason and talk about a possible way to achieve this goal. Another question that we view important but currently have no answer to is whether the $\mathcal{O}(T^{\frac{1-\mathfrak{p}}{3\mathfrak{p}-2}})$ rate is still achievable when $\mathfrak{p}$ is unknown, which we leave as an interesting direction to be explored in the future. Moreover, the reader may find that the rate in Theorem 3.6 loses the adaptivity on $\sigma_0$ since it can only guarantee the $\mathcal{O}(T^{-\frac{1}{4}})$ convergence instead of the optimal $\mathcal{O}(T^{-\frac{1}{2}})$ rate in the noiseless case. We remark that this is a common phenomenon for optimization algorithms when oblivious to the level of noise. For example, it is well-known that SGD suffers the same issue even under the finite variance assumption but when $\sigma_0$ is unknown. The last thing we have to mention is that, unfortunately, the good property of not needing the tail index $\mathfrak{p}$ may fail when considering $\sigma_1 > 0$ in the generalized heavy-tailed assumption. Precisely speaking, we can only prove that there exists a constant threshold $\sigma_1^* > 0$ such that $B$ can always be set to 1 if $\sigma_1 \leq \sigma_1^*$ and $B$ has to be chosen based on $\mathfrak{p}$ when $\sigma_1 > \sigma_1^*$. The details can be found in Theorem D.2 in the appendix.

In summary, we exhibit the first convergence result under heavy-tailed noises when only partial information about the problem is available. Particularly, under the classical heavy-tailed noises, we show how to guarantee convergence even if the tail index $\mathfrak{p}$ is unknown.

## 4 HOW GRADIENT NORMALIZATION WORKS

In this section, we will explain how gradient normalization works under heavy-tailed noises by both intuitive discussion and theoretical analysis. Moreover, to keep the analysis simple and better compare the difference between Batched NSGDM and Clipped SGD studied previously, we will focus on the classical $L_0$-smooth case (i.e., take $L_1 = 0$ in Assumption 2.2) under finite $\mathfrak{p}$-th moment noises (i.e., take $\sigma_1 = 0$ in Assumption 2.4) to align with prior works. Due to limited space, the missing proofs of presented lemmas (and their full version) are deferred in the appendix.

### 4.1 WHAT DOES GRADIENT CLIPPING DO?

Before analyzing Algorithm 1, let us first recap the update rule of Clipped SGD:

$$\boldsymbol{x}_{t+1} = \boldsymbol{x}_t - \eta_t \hat{\boldsymbol{g}}_t, \quad \hat{\boldsymbol{g}}_t \triangleq \min\left\{1, \frac{\tau_t}{\|\boldsymbol{g}_t\|}\right\}\boldsymbol{g}_t, \tag{1}$$

where $\hat{\boldsymbol{g}}_t$ is the clipped gradient and $\tau_t > 0$ is known as the clipping magnitude playing a critical role in the convergence. Especially, (1) can recover SGD by setting $\tau_t = +\infty$.

We provide a simple analysis here to illustrate how gradient clipping helps Clipped SGD to converge in expectation. First, by the well-known smoothness inequality, i.e., Lemma 2.5 with $L_1 = 0$, we know

$$
\begin{aligned}
F(\boldsymbol{x}_{t+1}) &\le F(\boldsymbol{x}_t) + \langle \nabla F(\boldsymbol{x}_t), \boldsymbol{x}_{t+1} - \boldsymbol{x}_t \rangle + \frac{L_0}{2} \|\boldsymbol{x}_{t+1} - \boldsymbol{x}_t\|^2 \\
&\overset{(1)}{=} F(\boldsymbol{x}_t) - \eta_t \langle \nabla F(\boldsymbol{x}_t), \hat{\boldsymbol{g}}_t \rangle + \frac{\eta_t^2 L_0}{2} \|\hat{\boldsymbol{g}}_t\|^2 \\
&= F(\boldsymbol{x}_t) - \left( \eta_t - \frac{\eta_t^2 L_0}{2} \right) \|\nabla F(\boldsymbol{x}_t)\|^2 - (\eta_t - \eta_t^2 L_0) \langle \nabla F(\boldsymbol{x}_t), \boldsymbol{\epsilon}_t \rangle + \frac{\eta_t^2 L_0}{2} \|\boldsymbol{\epsilon}_t\|^2,
\end{aligned}
\tag{2}
$$

where $\boldsymbol{\epsilon}_t \triangleq \hat{\boldsymbol{g}}_t - \nabla F(\boldsymbol{x}_t)$. Following the existing literature (e.g., Cutkosky & Mehta (2021); Liu et al. (2023b)), we next decompose $\boldsymbol{\epsilon}_t$ into $\boldsymbol{\epsilon}_t = \boldsymbol{\epsilon}_t^u + \boldsymbol{\epsilon}_t^b$, where $\boldsymbol{\epsilon}_t^u \triangleq \hat{\boldsymbol{g}}_t - \mathbb{E}[\hat{\boldsymbol{g}}_t \mid \mathcal{F}_{t-1}]$ and $\boldsymbol{\epsilon}_t^b \triangleq \mathbb{E}[\hat{\boldsymbol{g}}_t \mid \mathcal{F}_{t-1}] - \nabla F(\boldsymbol{x}_t)$. By noticing $\mathbb{E}[\langle \nabla F(\boldsymbol{x}_t), \boldsymbol{\epsilon}_t^u \rangle] = \mathbb{E}[\langle \boldsymbol{\epsilon}_t^b, \boldsymbol{\epsilon}_t^u \rangle] = 0$, we thus have

$$
\begin{aligned}
\mathbb{E}[F(\boldsymbol{x}_{t+1})] \le{}& \mathbb{E}[F(\boldsymbol{x}_t)] - \left( \eta_t - \frac{\eta_t^2 L_0}{2} \right) \mathbb{E}\left[ \|\nabla F(\boldsymbol{x}_t)\|^2 \right] \\
&+ (\eta_t - \eta_t^2 L_0) \mathbb{E}\left[ \langle \nabla F(\boldsymbol{x}_t), -\boldsymbol{\epsilon}_t^b \rangle \right] + \frac{\eta_t^2 L_0}{2} \mathbb{E}\left[ \|\boldsymbol{\epsilon}_t^u\|^2 + \|\boldsymbol{\epsilon}_t^b\|^2 \right] \\
\overset{\text{if } \eta_t \le 1/L_0}{\le}{}& \mathbb{E}[F(\boldsymbol{x}_t)] - \frac{\eta_t}{2} \mathbb{E}\left[ \|\nabla F(\boldsymbol{x}_t)\|^2 \right] + \frac{\eta_t^2 L_0}{2} \mathbb{E}\left[ \|\boldsymbol{\epsilon}_t^u\|^2 \right] + \frac{\eta_t}{2} \mathbb{E}\left[ \|\boldsymbol{\epsilon}_t^b\|^2 \right],
\end{aligned}
\tag{3}
$$

where the last step is by $\mathbb{E}\left[ \langle \nabla F(\boldsymbol{x}_t), -\boldsymbol{\epsilon}_t^b \rangle \right] \le \frac{\mathbb{E}[\|\nabla F(\boldsymbol{x}_t)\|^2] + \mathbb{E}[\|\boldsymbol{\epsilon}_t^b\|^2]}{2}$ and $\eta_t - \eta_t^2 L_0 \ge 0$.

- Now let us check what will happen for SGD if without clipping, i.e., taking $\tau_t = +\infty$. In this case, (3) recovers the classical one-step inequality for SGD by observing $\boldsymbol{\epsilon}_t^u = \boldsymbol{\xi}_t \triangleq \boldsymbol{g}_t - \nabla F(\boldsymbol{x}_t)$ and $\boldsymbol{\epsilon}_t^b = \boldsymbol{0}$ now, which also reveals why SGD may fail to converge because $\mathbb{E}\left[ \|\boldsymbol{\epsilon}_t^u\|^2 \right] = \mathbb{E}\left[ \|\boldsymbol{\xi}_t\|^2 \right]$ could be $+\infty$ under Assumption 2.4.

- In contrast, loosely speaking, setting $\tau_t < +\infty$ would ensure that both $\mathbb{E}\left[ \|\boldsymbol{\epsilon}_t^u\|^2 \right]$ and $\mathbb{E}\left[ \|\boldsymbol{\epsilon}_t^b\|^2 \right]$ are finite, whose upper bounds could be further controlled by picking $\tau_t$ properly. Finally, Clipped SGD would converge under carefully designed $\eta_t$ and $\tau_t$.

From the above comparison (which though is not strictly rigorous), one can intuitively think that the key thing done by gradient clipping is making the second moment of the error term $\mathbb{E}\left[ \|\boldsymbol{\epsilon}_t\|^2 \right]$ in (2) be bounded even when the noise $\boldsymbol{\xi}_t$ only has a finite $\mathfrak{p}$-th moment.

## 4.2 WHAT DOES GRADIENT NORMALIZATION DO?

By the discussion in the previous subsection, we can see that gradient clipping is used to control the second moment of the error term. A natural thought could be that we may avoid gradient clipping if the error term $\|\boldsymbol{\epsilon}_t\|^2$ in (2) is in a lower order, for example, say $\|\boldsymbol{\epsilon}_t\|^{\mathfrak{p}}$ which can be bounded in expectation directly without clipping. However, because $\mathfrak{p}$ is not necessarily known, we could aim to decrease the order of $\|\boldsymbol{\epsilon}_t\|$ from 2 to 1, i.e., the extreme case.

The above thought experiment may immediately help the reader who is familiar with the optimization literature recall the gradient normalization technique, in the analysis of which first-order terms always show up. As such, it is reasonable to expect that Batched NSGDM can converge under heavy-tailed noises even without knowing $\mathfrak{p}$ due to gradient normalization.

From now on, we start analyzing Algorithm 1 rigorously and formally showing how gradient normalization overcomes heavy-tailed noises.

**Lemma 4.1.** *Under Assumptions 2.1 and 2.2 (with $L_1 = 0$), let $\Delta_1 \triangleq F(\boldsymbol{x}_1) - F_*$, then Algorithm 1 guarantees*

$$
\sum_{t=1}^T \eta_t \mathbb{E}[\|\nabla F(\boldsymbol{x}_t)\|] \le \Delta_1 + \frac{L_0 \sum_{t=1}^T \eta_t^2}{2} + \sum_{t=1}^T 2 \eta_t \mathbb{E}[\|\boldsymbol{\epsilon}_t\|],
\tag{4}
$$

where $\epsilon_t \triangleq \boldsymbol{m}_t - \nabla F(\boldsymbol{x}_t), \forall t \in [T]$.

Lemma 4.1 provides a basic inequality used in the analysis of Batched NSGDM. We remark that this inequality is not new and has been proved many times before, even when $L_1 > 0$ (e.g., see Cutkosky & Mehta (2020); Jin et al. (2021)). But for completeness, the proof is provided in Appendix D.

As mentioned earlier, the first moment $\mathbb{E}[\|\epsilon_t\|]$ is exactly what we want, which we hope could be finite even under Assumption 2.4. To bound this term, we first recall the following widely used decomposition when studying gradient normalization (Cutkosky & Mehta, 2020; 2021; Jin et al., 2021; Liu et al., 2023b).

**Lemma 4.2.** *Algorithm 1 guarantees*

$$\epsilon_t = \beta_{1:t}\epsilon_0 + \sum_{s=1}^{t} \beta_{s:t}\boldsymbol{D}_s + \sum_{s=1}^{t}(1 - \beta_s)\beta_{s+1:t}\boldsymbol{\xi}_s, \forall t \in [T], \tag{5}$$

*where*

$$\epsilon_0 \triangleq \boldsymbol{g}_1 - \nabla F(\boldsymbol{x}_1), \quad \boldsymbol{D}_t \triangleq \begin{cases} \nabla F(\boldsymbol{x}_{t-1}) - \nabla F(\boldsymbol{x}_t) & 2 \le t \le T \\ \boldsymbol{0} & t = 1 \end{cases}, \quad \boldsymbol{\xi}_t \triangleq \boldsymbol{g}_t - \nabla F(\boldsymbol{x}_t).$$

*Proof.* By the definition of $\epsilon_t$ when $t \ge 2$,

$$\begin{aligned} \epsilon_t = \boldsymbol{m}_t - \nabla F(\boldsymbol{x}_t) &= \beta_t \boldsymbol{m}_{t-1} + (1 - \beta_t)\boldsymbol{g}_t - \nabla F(\boldsymbol{x}_t) \\ &= \beta_t(\boldsymbol{m}_{t-1} - \nabla F(\boldsymbol{x}_{t-1})) + \beta_t(\nabla F(\boldsymbol{x}_{t-1}) - \nabla F(\boldsymbol{x}_t)) + (1 - \beta_t)(\boldsymbol{g}_t - \nabla F(\boldsymbol{x}_t)) \\ &= \beta_t \epsilon_{t-1} + \beta_t \boldsymbol{D}_t + (1 - \beta_t)\boldsymbol{\xi}_t. \end{aligned}$$

One can verify that the above equation also holds when $t = 1$ under our notations (recall $\boldsymbol{m}_0 = \boldsymbol{g}_1$ in Algorithm 1). Unrolling the equation recursively, we can finally obtain the desired result. □

After plugging (5) into (4), as one can imagine, our remaining task is to upper bound $\mathbb{E}[\|\epsilon_0\|]$, $\mathbb{E}\left[\left\|\sum_{s=1}^{t}\beta_{s:t}\boldsymbol{D}_s\right\|\right]$ and $\mathbb{E}\left[\left\|\sum_{s=1}^{t}(1-\beta_s)\beta_{s+1:t}\boldsymbol{\xi}_s\right\|\right]$. For simplicity, we consider the batch size $B = 1$ in the following.

First, note that $\mathbb{E}\left[\left\|\sum_{s=1}^{t}\beta_{s:t}\boldsymbol{D}_s\right\|\right] \le \sum_{s=1}^{t}\beta_{s:t}\mathbb{E}[\|\boldsymbol{D}_s\|]$ and $\|\boldsymbol{D}_s\|$ can be bounded by $L_0$-smoothness easily (even under $(L_0, L_1)$-smoothness), hence we skip the calculation here. Next, when $B = 1$, one can use Hölder's inequality to bound $\mathbb{E}[\|\epsilon_0\|] \le \left(\mathbb{E}[\|\epsilon_0\|^{\mathfrak{p}}]\right)^{\frac{1}{\mathfrak{p}}} \le \sigma_0$ where the last step is by Assumption 2.4 when $\sigma_1 = 0$. For the left term $\mathbb{E}\left[\left\|\sum_{s=1}^{t}(1-\beta_s)\beta_{s+1:t}\boldsymbol{\xi}_s\right\|\right]$:

- When $\mathfrak{p} = 2$, prior works like Cutkosky & Mehta (2020) invoke Hölder's inequality to have

$$\begin{aligned} \mathbb{E}\left[\left\|\sum_{s=1}^{t}(1-\beta_s)\beta_{s+1:t}\boldsymbol{\xi}_s\right\|\right] &\le \sqrt{\mathbb{E}\left[\left\|\sum_{s=1}^{t}(1-\beta_s)\beta_{s+1:t}\boldsymbol{\xi}_s\right\|^2\right]} \\ &\le \sqrt{\sum_{s=1}^{t}((1-\beta_s)\beta_{s+1:t}\sigma_0)^2}, \end{aligned}$$

where the last step is due to $\mathbb{E}[\langle\boldsymbol{\xi}_s, \boldsymbol{\xi}_t\rangle] = 0$ for every cross term and $\mathbb{E}\left[\|\boldsymbol{\xi}_s\|^2\right] \le \sigma_0^2$.

- Naturally, when noises only have finite $\mathfrak{p}$-th moments, one may want to apply Hölder's inequality in the following form,

$$\mathbb{E}\left[\left\|\sum_{s=1}^{t}(1-\beta_s)\beta_{s+1:t}\boldsymbol{\xi}_s\right\|\right] \le \left(\mathbb{E}\left[\left\|\sum_{s=1}^{t}(1-\beta_s)\beta_{s+1:t}\boldsymbol{\xi}_s\right\|^{\mathfrak{p}}\right]\right)^{\frac{1}{\mathfrak{p}}}.$$

However, a critical issue here is that $\|\cdot\|^{\mathfrak{p}}$ cannot be expanded like $\|\cdot\|^2$ to make the cross term $\langle\boldsymbol{\xi}_s, \boldsymbol{\xi}_t\rangle$ show up, which fails the analysis.

As such, how to establish a meaningful bound on $\mathbb{E}\left[\left\|\sum_{s=1}^{t}(1-\beta_s)\beta_{s+1:t}\boldsymbol{\xi}_s\right\|\right]$ is the novel part of our proofs, which essentially differs from the existing analysis when $\mathfrak{p}=2$.

Fix $t \in [T]$, to ease the notation, we denote by $\boldsymbol{v}_s \triangleq (1-\beta_s)\beta_{s+1:t}\boldsymbol{\xi}_s, \forall s \in [t]$. Hence, our goal can be summarized as bounding the norm of $\sum_{s=1}^{t} \boldsymbol{v}_s$, where $\boldsymbol{v}_s$ is a vector-valued martingale difference sequence (MDS). To do so, we introduce the following inequality, which is the core of our analysis.

**Lemma 4.3.** *Given a sequence of integrable random vectors $\boldsymbol{v}_t \in \mathbb{R}^d, \forall t \in \mathbb{N}$ such that $\mathbb{E}[\boldsymbol{v}_t \mid \mathcal{F}_{t-1}] = \mathbf{0}$ where $\mathcal{F}_t \triangleq \sigma(\boldsymbol{v}_1, \cdots, \boldsymbol{v}_t)$ is the natural filtration, then for any $\mathfrak{p} \in [1, 2]$, there is*

$$\mathbb{E}\left[\left\|\sum_{t=1}^{T} \boldsymbol{v}_t\right\|\right] \leq 2\sqrt{2}\mathbb{E}\left[\left(\sum_{t=1}^{T}\|\boldsymbol{v}_t\|^{\mathfrak{p}}\right)^{\frac{1}{\mathfrak{p}}}\right], \forall T \in \mathbb{N}. \tag{6}$$

At first glance, (6) seems wrong because it provides $\mathbb{E}\left[\left\|\sum_{t=1}^{T} \boldsymbol{v}_t\right\|\right] \leq 2\sqrt{2}\mathbb{E}\left[\sqrt{\sum_{t=1}^{T}\|\boldsymbol{v}_t\|^2}\right]$ by taking $\mathfrak{p}=2$. However, one may only expect $\mathbb{E}\left[\left\|\sum_{t=1}^{T}\boldsymbol{v}_t\right\|\right] \leq \sqrt{\sum_{t=1}^{T}\mathbb{E}\left[\|\boldsymbol{v}_t\|^2\right]}$ to hold. So why is (6) true? Intuitively, this is because (6) is only stated for the MDS in contrast to Hölder's inequality being able to apply to any sequence.

To let the reader believe Lemma 4.3 is correct, we first consider the case of $d=1$ and recall the famous Burkholder-Davis-Gundy (BDG) inequality.

**Lemma 4.4.** *(Burkholder-Davis-Gundy Inequality (Burkholder, 1966; Burkholder & Gundy, 1970; Davis, 1970), simplified version) Given a discrete martingale $X_t \in \mathbb{R}$ with $X_0 = 0$, then there exists a constant $C_1 > 0$ such that*

$$\mathbb{E}\left[\max_{t\in[T]}|X_t|\right] \leq C_1\mathbb{E}\left[\sqrt{\sum_{t=1}^{T}(X_t - X_{t-1})^2}\right], \forall T \in \mathbb{N}.$$

Let $X_t \triangleq \sum_{s=1}^{t}\boldsymbol{v}_s$, BDG inequality immediately implies (6) under $d=1$ and $\mathfrak{p}=2$ (up to a constant) due to

$$\mathbb{E}\left[\left|\sum_{t=1}^{T}\boldsymbol{v}_t\right|\right] = \mathbb{E}[|X_T|] \leq \mathbb{E}\left[\max_{t\in[T]}|X_t|\right] \leq C_1\mathbb{E}\left[\sqrt{\sum_{t=1}^{T}(X_t - X_{t-1})^2}\right] = C_1\mathbb{E}\left[\sqrt{\sum_{t=1}^{T}|\boldsymbol{v}_t|^2}\right].$$

One more step, by noticing $\|\cdot\| \leq \|\cdot\|_{\mathfrak{p}}$ for $\mathfrak{p} \in [1,2]$, Lemma 4.3 thereby holds when $d=1$ by

$$\mathbb{E}\left[\left|\sum_{t=1}^{T}\boldsymbol{v}_t\right|\right] \leq C_1\mathbb{E}\left[\sqrt{\sum_{t=1}^{T}|\boldsymbol{v}_t|^2}\right] \leq C_1\mathbb{E}\left[\left(\sum_{t=1}^{T}|\boldsymbol{v}_t|^{\mathfrak{p}}\right)^{\frac{1}{\mathfrak{p}}}\right].$$

Though we have applied BDG inequality to prove Lemma 4.3 for $d=1$, extending the above analysis to the high-dimensional case is not obvious and could be non-trivial.

Here, inspired by Rakhlin & Sridharan (2017), we will provide a simple proof of Lemma 4.3 via the regret analysis from *online learning*. Specifically, we will prove the regret bound of the famous AdaGrad algorithm (McMahan & Streeter, 2010; Duchi et al., 2011) implies Lemma 4.3, which is kindly surprising (at least in our opinion) and shows the impressive power of online learning. Due to space limitations, the proof of Lemma 4.3 is deferred to Appendix C.

Before moving on, we make two comments on Lemma 4.3. First, as one can see, Lemma 4.3 holds for any $\mathfrak{p} \in [1, 2]$ meaning that this analysis is automatically adaptive to the tail index $\mathfrak{p}$. Next, one may wonder why we keep the expectation outside on the R.H.S. of (6) instead of putting it inside by Hölder's inequality. This is because under the full version of Assumption 2.4 (i.e., $\sigma_1 > 0$), making expectations inside may fail the analysis. For details, we refer the reader to the proof of Lemma D.5 in the appendix.

Now, we can apply Lemma 4.3 to bound $\mathbb{E}\left[\left\|\sum_{s=1}^{t}(1-\beta_s)\beta_{s+1:t}\boldsymbol{\xi}_s\right\|\right]$ under Assumption 2.4 with $\sigma_1 = 0$ and then obtain the following inequality for $\mathbb{E}\left[\|\boldsymbol{\epsilon}_t\|\right]$ by combining the previous bounds on $\mathbb{E}\left[\|\boldsymbol{\epsilon}_0\|\right]$ and $\mathbb{E}\left[\left\|\sum_{s=1}^{t}\beta_{s:t}\boldsymbol{D}_s\right\|\right]$.

**Lemma 4.5.** *Under Assumptions 2.2 (with $L_1 = 0$), 2.3 (with $B = 1$) and 2.4 (with $\sigma_1 = 0$), then Algorithm 1 guarantees*

$$\mathbb{E}\left[\|\boldsymbol{\epsilon}_t\|\right] \le 2\sqrt{2}\left[\beta_{1:t}\sigma_0 + \left(\sum_{s=1}^{t}(1-\beta_s)^{\mathfrak{p}}(\beta_{s+1:t})^{\mathfrak{p}}\right)^{\frac{1}{\mathfrak{p}}}\sigma_0\right] + \sum_{s=2}^{t}\beta_{s:t}L_0\eta_{s-1}, \forall t \in [T].$$

The full version of the above result, Lemma D.5, that works for any $L_1, B, \sigma_1$ can be found in Appendix D, which requires extra effort as mentioned earlier.

Finally, equipped with Lemmas 4.1 and 4.5, we are able to prove the convergence of Batched NS-GDM under the classical smooth condition and heavy-tailed noises by plugging in the stepsize and momentum parameter introduced in Theorems 3.2 and 3.6, respectively.

Before ending this section, we briefly talk about the intuition behind the rate $\mathcal{O}(T^{\frac{1-\mathfrak{p}}{2\mathfrak{p}}})$ achieved when the tail index $\mathfrak{p}$ is unknown, i.e., Theorem 3.6. Actually, we take a quite simple strategy: setting the stepsize and the momentum parameter while pretending the tail index $\mathfrak{p}$ to be 2. Amazingly, this straightforward policy is already enough to guarantee convergence even if there is no prior information on $\mathfrak{p}$.

## 5 Conclusion and Future Work

In this work, we present the first optimal expected convergence result under heavy-tailed noises but without gradient clipping, which is instead achieved by gradient normalization. More specifically, we study the existing Batched NSGDM algorithm and prove it converges in expectation at an optimal $\mathcal{O}(T^{\frac{1-\mathfrak{p}}{3\mathfrak{p}-2}})$ rate. Additionally, the order of problem-dependent parameters in our upper bound is also the first to be tight as indicated by a newly matched lower bound improved from the prior work. One step further, we initiate the study of convergence under heavy-tailed noises but without knowing the tail index $\mathfrak{p}$ and then obtain the first provable $\mathcal{O}(T^{\frac{1-\mathfrak{p}}{2\mathfrak{p}}})$ rate. Thus, our work suggests gradient normalization is a powerful tool for dealing with heavy-tailed noises, which we believe will bring new insights into the optimization community and open potential ways for future algorithm design.

However, there still remain some directions worth exploring, and we list three specific topics here:

**Minimax rate for unknown tail index $\mathfrak{p}$.** As previously discussed, to achieve the minimax $\Theta(T^{\frac{1-\mathfrak{p}}{3\mathfrak{p}-2}})$ rate under heavy-tailed noises, all optimal algorithms so far require knowing the tail index $\mathfrak{p}$. Thus, it would be interesting to consider the optimal upper/lower bound of the convergence rate when $\mathfrak{p}$ is unknown. We provide two concrete problems here and hope them being addressed in the future: 1. When lacking any prior information on $\mathfrak{p}$, is it possible to find an algorithm that can improve our new $\mathcal{O}(T^{\frac{1-\mathfrak{p}}{2\mathfrak{p}}})$ upper bound to the best-known $\mathcal{O}(T^{\frac{1-\mathfrak{p}}{3\mathfrak{p}-2}})$ rate? 2. If the answer to the former question is negative, what is the corresponding lower bound when $\mathfrak{p}$ is unknown?

**Adaptive gradient methods.** Though we have established the first convergence result under heavy-tailed noises without gradient clipping, the Batched NSGDM algorithm we studied is not commonly used in practice. In comparison, the family of adaptive gradient methods (e.g., AdaGrad (McMahan & Streeter, 2010; Duchi et al., 2011), RMSprop (Tieleman et al., 2012), Adam (Kingma & Ba, 2014) and their variants) is more popular and has been widely implemented nowadays, especially when training neural networks. Surprisingly, their performances are still good even though the stochastic noises are empirically observed to be heavy-tailed. However, as far as we know, no rigorous theoretical justification has been established to show that adaptive gradient methods can converge under heavy-tailed noises. Hence, it is worth studying and trying to close this important gap between theory and practice.

**Time-varying choices.** Another potential extension is to study the time-varying stepsize and momentum parameter to make the algorithm more practical, which we believe is possible given our general lemmas.

ACKNOWLEDGMENTS

This work is supported by the NSF grant ECCS-2419564. We also thank the anonymous reviewers for their valuable comments.

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

## A  AN EXAMPLE FAILS THE EXISTING HEAVY-TAILED NOISES ASSUMPTION

In this section, we provide a simple one-dimensional case that violates the previously used finite $\mathfrak{p}$-th moment assumption but satisfies our new generalized heavy-tailed noises condition, Assumption 2.4. Extending the example to high dimensions is straightforward, which is left to the reader.

**Example A.1.** Let $F(\boldsymbol{x}) \triangleq \frac{1}{2}\mathbb{E}_{a,b}\left[(a\boldsymbol{x}-b)^2\right]$ where $\boldsymbol{x} \in \mathbb{R}$, $a \triangleq \mathrm{Bernoulli}(q)$ for some $q \in (0,1)$, and $b \triangleq a\boldsymbol{x}_* + \omega$ where $\boldsymbol{x}_* \in \mathbb{R}$ is fixed and $\omega$ is a centered random variable being independent of $a$ and only has a finite $\mathfrak{p}$-th moment (i.e., $\mathbb{E}_\omega\left[|\omega|^\mathfrak{p}\right] \leq \sigma^\mathfrak{p}$ for some $\sigma \geq 0$). For the stochastic gradient $\boldsymbol{g}(\boldsymbol{x},a,b) \triangleq a^2\boldsymbol{x} - ab$ and the true gradient $\nabla F(\boldsymbol{x}) \triangleq \mathbb{E}_a\left[a^2\right]\boldsymbol{x} - \mathbb{E}_{a,b}\left[ab\right]$, we let the the noise be $\boldsymbol{\xi}(\boldsymbol{x}) \triangleq \boldsymbol{g}(\boldsymbol{x},a,b) - \nabla F(\boldsymbol{x})$, then

- Assumption 2.4 is failed for any pair $(\sigma_0, 0)$ where $\sigma_0 \geq 0$ can be arbitrary;

- Assumption 2.4 is satisfied for a certain pair $(\sigma_0, \sigma_1)$ where $\sigma_1 > 0$.

*Proof.* Note that $\boldsymbol{g}(\boldsymbol{x},a,b)$ and $\nabla F(\boldsymbol{x})$ can be simplified into

$$\boldsymbol{g}(\boldsymbol{x},a,b) = a^2(\boldsymbol{x}-\boldsymbol{x}_*) - a\omega, \quad \nabla F(\boldsymbol{x}) = q(\boldsymbol{x}-\boldsymbol{x}_*).$$

Thus, we know

$$\boldsymbol{\xi}(\boldsymbol{x}) = (a^2 - q)(\boldsymbol{x}-\boldsymbol{x}_*) - a\omega$$

$$\Rightarrow \mathbb{E}_{a,b}\left[|\boldsymbol{\xi}(\boldsymbol{x})|^\mathfrak{p}\right] = \mathbb{E}_{a,\omega}\left[\left|(a^2-q)(\boldsymbol{x}-\boldsymbol{x}_*)-a\omega\right|^\mathfrak{p}\right]$$

$$= (1-q)q^\mathfrak{p}\left|\boldsymbol{x}-\boldsymbol{x}_*\right|^\mathfrak{p} + q\mathbb{E}_\omega\left[\left|(1-q)(\boldsymbol{x}-\boldsymbol{x}_*)-\omega\right|^\mathfrak{p}\right].$$

On the one side, we have

$$\mathbb{E}_{a,b}\left[|\boldsymbol{\xi}(\boldsymbol{x})|^\mathfrak{p}\right] \geq (1-q)q^\mathfrak{p}\left|\boldsymbol{x}-\boldsymbol{x}_*\right|^\mathfrak{p} \overset{\boldsymbol{x}\to\pm\infty}{\longrightarrow} +\infty,$$

Therefore, $\mathbb{E}_{a,b}\left[|\boldsymbol{\xi}(\boldsymbol{x})|^\mathfrak{p}\right] \leq \sigma_0^\mathfrak{p}$ cannot hold for any $\sigma_0 \geq 0$.

On the other side, we observe

$$\mathbb{E}_\omega\left[\left|(1-q)(\boldsymbol{x}-\boldsymbol{x}_*)-\omega\right|^\mathfrak{p}\right] \leq \mathbb{E}_\omega\left[\left|(1-q)\left|\boldsymbol{x}-\boldsymbol{x}_*\right| + |\omega|\right|^\mathfrak{p}\right]$$

$$\leq 2^{\mathfrak{p}-1}(1-q)^\mathfrak{p}\left|\boldsymbol{x}-\boldsymbol{x}_*\right|^\mathfrak{p} + 2^{\mathfrak{p}-1}\mathbb{E}_\omega\left[|\omega|^\mathfrak{p}\right]$$

$$\leq 2^{\mathfrak{p}-1}(1-q)^\mathfrak{p}\left|\boldsymbol{x}-\boldsymbol{x}_*\right|^\mathfrak{p} + 2^{\mathfrak{p}-1}\sigma^\mathfrak{p}.$$

Hence,

$$\mathbb{E}_{a,b}\left[|\boldsymbol{\xi}(\boldsymbol{x})|^\mathfrak{p}\right] \leq 2^{\mathfrak{p}-1}q\sigma^\mathfrak{p} + \left[(1-q)q^\mathfrak{p} + 2^{\mathfrak{p}-1}q(1-q)^\mathfrak{p}\right]\left|\boldsymbol{x}-\boldsymbol{x}_*\right|^\mathfrak{p}$$

$$= 2^{\mathfrak{p}-1}q\sigma^\mathfrak{p} + \left[1-q+2^{\mathfrak{p}-1}q^{1-\mathfrak{p}}(1-q)^\mathfrak{p}\right]\left|\nabla F(\boldsymbol{x})\right|^\mathfrak{p}.$$

So Assumption 2.4 is satisfied for $\sigma_0^\mathfrak{p} \triangleq 2^{\mathfrak{p}-1}q\sigma^\mathfrak{p}$ and $\sigma_1^\mathfrak{p} \triangleq 1-q+2^{\mathfrak{p}-1}q^{1-\mathfrak{p}}(1-q)^\mathfrak{p}$. □

## B  FURTHER DISCUSSION ON $L_1$ IN THEOREM 3.6

One may think knowing $L_1$ is necessary under the present version of $(L_0, L_1)$-smoothness (i.e., Assumption 2.2). It is because the current form of $(L_0, L_1)$-smoothness can be only invoked under the hard constraint $\|\boldsymbol{x} - \boldsymbol{y}\| \leq \frac{1}{L_1}$. As such, regardless of whichever two points $\boldsymbol{x}_s$ and $\boldsymbol{x}_t$ we want to apply to the descent inequality in Lemma D.4 (which serves as the cornerstone in the whole proof), they have to satisfy $\|\boldsymbol{x}_s - \boldsymbol{x}_t\| \leq \frac{1}{L_1}$. In other words, one has to know $L_1$ to set the stepsize to ensure Lemma D.4 can be used. Moreover, even if when $\mathfrak{p} = 2$ or under a weaker condition on noises (e.g., almost surely bounded noises), prior works under exactly the same form of Assumption 2.2 require the value of $L_1$ (e.g., see Jin et al. (2021); Crawshaw et al. (2022)), even for the famous adaptive method – AdaGrad (Faw et al., 2023; Wang et al., 2023; Hong & Lin, 2024) and Adam (Wang et al., 2024).

However, this turns out to be too pessimistic. To see why, we first claim that Assumption 2.2 is essentially equivalent to the following condition (up to redefining the pair $(L_0, L_1)$)

$$\|\nabla F(\boldsymbol{x}) - \nabla F(\boldsymbol{y})\| \leq (L_0 + L_1\|\nabla F(\boldsymbol{x})\|)\|\boldsymbol{x} - \boldsymbol{y}\|\exp(L_1\|\boldsymbol{x}-\boldsymbol{y}\|), \forall \boldsymbol{x}, \boldsymbol{y} \in \mathbb{R}^d. \quad (7)$$

*Remark* B.1. (7) is also known to be equivalent to the $\mathcal{L}^*_{\text{sym}}(1)$ function class. See Definition 3 and Proposition 1.2 in Chen et al. (2023) for details.

This means that for any $\boldsymbol{x}, \boldsymbol{y} \in \mathbb{R}^d$, we can still bound the difference between their gradients even if the distance between them is larger than $\frac{1}{L_1}$, but incurring an extra multiplicative term as a penalty. As such, it is possible to combine the dynamic stepsize and time-varying momentum, i.e., taking $\eta_t = \eta t^{-a}$ and $\beta_t = \beta t^{-b}$ for some $\eta, \beta, a, b > 0$, to get rid of knowing $L_1$ since we can control the size of gradient norm during the warm-up period (i.e., the time before $\eta_t = \mathcal{O}(L_1^{-1})$) and then follow a similar proof in our paper when $\eta_t = \mathcal{O}(L_1^{-1})$ to obtain the desired result. Whereas, as a trade-off, one has to incur an exponential dependence on $L_1$ in the final convergence rate as pointed out by Hübler et al. (2024).

Lastly, let us prove this important equivalence.

- (7) implies Assumption 2.2. This is the easy case. Suppose (7) holds, then for any $\boldsymbol{x}, \boldsymbol{y} \in \mathbb{R}^d$ satisfying $\|\boldsymbol{x} - \boldsymbol{y}\| \leq \frac{1}{L_1}$, there is

$$\|\nabla F(\boldsymbol{x}) - \nabla F(\boldsymbol{y})\| \leq (eL_0 + eL_1 \|\nabla F(\boldsymbol{x})\|) \|\boldsymbol{x} - \boldsymbol{y}\|,$$

  which implies $F$ is $(eL_0, eL_1)$-smooth in the sense of Assumption 2.2.

- Assumption 2.2 implies (7)[4]. This direction is not obvious, whose proof in the following is inspired by Chen et al. (2023). Given $\boldsymbol{x}, \boldsymbol{y} \in \mathbb{R}^d$, we define the linear interpolation $\boldsymbol{z}_{\boldsymbol{x},\boldsymbol{y}}(t) \triangleq t\boldsymbol{x} + (1-t)\boldsymbol{y}, \forall t \in [0,1]$. Note that for any $n \in \mathbb{N}$ satisfying $n \geq L_1 \|\boldsymbol{x} - \boldsymbol{y}\|$ and $i \in [n]$, there is

$$\|\boldsymbol{z}_{\boldsymbol{x},\boldsymbol{y}}(i/n) - \boldsymbol{z}_{\boldsymbol{x},\boldsymbol{y}}(i-1/n)\| = \frac{\|\boldsymbol{x} - \boldsymbol{y}\|}{n} \leq \frac{1}{L_1}.$$

  Therefore, we can use Assumption 2.2 to control $\|\nabla F(\boldsymbol{z}_{\boldsymbol{x},\boldsymbol{y}}(i/n)) - \nabla F(\boldsymbol{z}_{\boldsymbol{x},\boldsymbol{y}}(i-1/n))\|$ and obtain the following bound whenever $n \geq L_1 \|\boldsymbol{x} - \boldsymbol{y}\|$,

$$\|\nabla F(\boldsymbol{x}) - \nabla F(\boldsymbol{y})\|$$
$$= \left\| \sum_{i=1}^{n} \nabla F(\boldsymbol{z}_{\boldsymbol{x},\boldsymbol{y}}(i/n)) - \nabla F(\boldsymbol{z}_{\boldsymbol{x},\boldsymbol{y}}(i-1/n)) \right\| \leq \sum_{i=1}^{n} \|\nabla F(\boldsymbol{z}_{\boldsymbol{x},\boldsymbol{y}}(i/n)) - \nabla F(\boldsymbol{z}_{\boldsymbol{x},\boldsymbol{y}}(i-1/n))\|$$
$$\leq \frac{\|\boldsymbol{x} - \boldsymbol{y}\|}{n} \sum_{i=1}^{n} L_0 + L_1 \|\nabla F(\boldsymbol{z}_{\boldsymbol{x},\boldsymbol{y}}(i-1/n))\| = \frac{\|\boldsymbol{x} - \boldsymbol{y}\|}{n} \sum_{i=1}^{n} h(i-1/n), \quad (8)$$

  where $h(t) \triangleq L_0 + L_1 \|\nabla F(\boldsymbol{z}_{\boldsymbol{x},\boldsymbol{y}}(t))\|, \forall t \in [0,1]$. We notice that $h$ is continuous because for any given $t \in (0,1)$ and $\varepsilon > 0$, for $s \in [0,1]$ satisfying $|s - t| \leq \frac{\min\{h(t),\varepsilon\}}{L_1 h(t)\|\boldsymbol{x}-\boldsymbol{y}\|}$, there is

$$|h(s) - h(t)| = L_1 |\|\nabla F(\boldsymbol{z}_{\boldsymbol{x},\boldsymbol{y}}(s))\| - \|\nabla F(\boldsymbol{z}_{\boldsymbol{x},\boldsymbol{y}}(t))\|| \leq L_1 \|\nabla F(\boldsymbol{z}_{\boldsymbol{x},\boldsymbol{y}}(s)) - \nabla F(\boldsymbol{z}_{\boldsymbol{x},\boldsymbol{y}}(t))\|$$
$$\overset{(a)}{\leq} L_1 h(t) \|\boldsymbol{z}_{\boldsymbol{x},\boldsymbol{y}}(s) - \boldsymbol{z}_{\boldsymbol{x},\boldsymbol{y}}(t)\| = L_1 h(t) \|\boldsymbol{x} - \boldsymbol{y}\| |s - t| \overset{(b)}{\leq} \varepsilon,$$

  where $(a)$ is due to $\|\boldsymbol{z}_{\boldsymbol{x},\boldsymbol{y}}(s) - \boldsymbol{z}_{\boldsymbol{x},\boldsymbol{y}}(t)\| = \|\boldsymbol{x} - \boldsymbol{y}\| |s - t| \leq \frac{1}{L_1}$ and Assumption 2.2, and $(b)$ holds by $|s - t| \leq \frac{\varepsilon}{L_1 h(t)\|\boldsymbol{x}-\boldsymbol{y}\|}$. Thus, $h$ is integrable on $[0,1]$, which implies

$$\|\nabla F(\boldsymbol{x}) - \nabla F(\boldsymbol{y})\| \overset{(8)}{\leq} \|\boldsymbol{x} - \boldsymbol{y}\| \lim_{n \to \infty} \frac{1}{n} \sum_{i=1}^{n} h(i-1/n) = \|\boldsymbol{x} - \boldsymbol{y}\| \int_0^1 h(t)\mathrm{d}t. \quad (9)$$

  By Lemma 2 and Proposition 1.2 in Chen et al. (2023), (9) is equivalent to (7).

## C  Proof of the Core Lemma 4.3 and Some Discussions

Before proving Lemma 4.3, we need the following technique tool, which is based on the regret analysis of the AdaGrad algorithm as mentioned in Subsection 4.2.

---

[4]We especially thank the anonymous Reviewer tQr8 for suggesting this direction during the discussion.

**Lemma C.1.** *(Based on Lemma 2 in Rakhlin & Sridharan (2017)) Given a sequence of vectors $\boldsymbol{v}_t \in \mathbb{R}^d, \forall t \in \mathbb{N}$, then there exists a sequence of vectors $\boldsymbol{w}_t \in \mathbb{R}^d$ such that $\|\boldsymbol{w}_t\| \leq 1$ and every $\boldsymbol{w}_t$ only depends on $\boldsymbol{v}_1$ to $\boldsymbol{v}_{t-1}$ satisfying*

$$\left\| \sum_{t=1}^{T} \boldsymbol{v}_t \right\| \leq 2\sqrt{2\sum_{t=1}^{T} \|\boldsymbol{v}_t\|^2 - \sum_{t=1}^{T} \langle \boldsymbol{v}_t, \boldsymbol{w}_t \rangle}, \forall T \in \mathbb{N}.$$

*Proof.* W.l.o.g., we assume $\|\boldsymbol{v}_1\| > 0$. Otherwise, we let $\tau \triangleq \operatorname{argmin} \{t \in \mathbb{N} : \|\boldsymbol{v}_t\| > 0\}$, set $\boldsymbol{w}_t \triangleq \boldsymbol{0} \in \mathbb{R}^d, \forall t \in [\tau - 1]$, and start the following proof at time $\tau$.

Let $\boldsymbol{w}_1 \triangleq \boldsymbol{0}$ and recursively define $\boldsymbol{w}_{t+1} \triangleq \Pi_{\mathbb{B}^d} (\boldsymbol{w}_t - \gamma_t \boldsymbol{v}_t)$ where $\Pi_{\mathbb{B}^d}$ denotes the Euclidean projection operator onto the unit ball $\mathbb{B}^d$ in $\mathbb{R}^d$ and $\gamma_t \triangleq \sqrt{\frac{2}{\sum_{s=1}^{t} \|\boldsymbol{v}_s\|^2}}$. Clearly, $\|\boldsymbol{w}_t\| \leq 1$ and $\boldsymbol{w}_t$ only depends on $\boldsymbol{v}_1$ to $\boldsymbol{v}_{t-1}$ by construction. Next, observe that

$$\boldsymbol{w}_{t+1} = \Pi_{\mathbb{B}^d} (\boldsymbol{w}_t - \gamma_t \boldsymbol{v}_t) = \operatorname{argmin}_{\boldsymbol{w} \in \mathbb{B}^d} \|\boldsymbol{w} - \boldsymbol{w}_t + \gamma_t \boldsymbol{v}_t\|^2,$$

which by the optimality condition implies for any $\boldsymbol{u} \in \mathbb{B}^d$,

$$\langle \boldsymbol{w}_{t+1} - \boldsymbol{w}_t + \gamma_t \boldsymbol{v}_t, \boldsymbol{w}_{t+1} - \boldsymbol{u} \rangle \leq 0$$

$$\Rightarrow \langle \boldsymbol{v}_t, \boldsymbol{w}_{t+1} - \boldsymbol{u} \rangle \leq \frac{\langle \boldsymbol{w}_t - \boldsymbol{w}_{t+1}, \boldsymbol{w}_{t+1} - \boldsymbol{u} \rangle}{\gamma_t}$$

$$= \frac{\|\boldsymbol{u} - \boldsymbol{w}_t\|^2 - \|\boldsymbol{u} - \boldsymbol{w}_{t+1}\|^2 - \|\boldsymbol{w}_t - \boldsymbol{w}_{t+1}\|^2}{2\gamma_t}$$

$$\Rightarrow \langle \boldsymbol{v}_t, \boldsymbol{w}_t - \boldsymbol{u} \rangle \leq \frac{\|\boldsymbol{u} - \boldsymbol{w}_t\|^2 - \|\boldsymbol{u} - \boldsymbol{w}_{t+1}\|^2 - \|\boldsymbol{w}_t - \boldsymbol{w}_{t+1}\|^2}{2\gamma_t} + \langle \boldsymbol{v}_t, \boldsymbol{w}_t - \boldsymbol{w}_{t+1} \rangle$$

$$\overset{(a)}{\leq} \frac{\|\boldsymbol{u} - \boldsymbol{w}_t\|^2 - \|\boldsymbol{u} - \boldsymbol{w}_{t+1}\|^2}{2\gamma_t} + \frac{\gamma_t \|\boldsymbol{v}_t\|^2}{2},$$

where $(a)$ is by the Arithmetic Mean-Geometric Mean inequality. Hence, for any $\boldsymbol{u} \in \mathbb{B}^d$ and $T \in \mathbb{N}$,

$$\sum_{t=1}^{T} \langle \boldsymbol{v}_t, \boldsymbol{w}_t - \boldsymbol{u} \rangle \leq \sum_{t=1}^{T} \frac{\|\boldsymbol{u} - \boldsymbol{w}_t\|^2 - \|\boldsymbol{u} - \boldsymbol{w}_{t+1}\|^2}{2\gamma_t} + \frac{\gamma_t \|\boldsymbol{v}_t\|^2}{2}$$

$$= \frac{\|\boldsymbol{u} - \boldsymbol{w}_1\|^2}{2\gamma_1} - \frac{\|\boldsymbol{u} - \boldsymbol{w}_{T+1}\|^2}{2\gamma_T} + \sum_{t=2}^{T} \frac{\|\boldsymbol{u} - \boldsymbol{w}_t\|^2}{2}\left(\frac{1}{\gamma_t} - \frac{1}{\gamma_{t-1}}\right) + \sum_{t=1}^{T} \frac{\gamma_t \|\boldsymbol{v}_t\|^2}{2}$$

$$\overset{(b)}{\leq} \frac{2}{\gamma_1} + \sum_{t=2}^{T} 2\left(\frac{1}{\gamma_t} - \frac{1}{\gamma_{t-1}}\right) + \sum_{t=1}^{T} \frac{\gamma_t \|\boldsymbol{v}_t\|^2}{2} = \frac{2}{\gamma_T} + \sum_{t=1}^{T} \frac{\gamma_t \|\boldsymbol{v}_t\|^2}{2}, \qquad (10)$$

where $(b)$ is due to $\frac{1}{\gamma_t} - \frac{1}{\gamma_{t-1}} \geq 0, \forall t \geq 2$ by the definition of $\gamma_t = \sqrt{\frac{2}{\sum_{s=1}^{t} \|\boldsymbol{v}_s\|^2}}$ and $\|\boldsymbol{u} - \boldsymbol{w}_t\| \leq 2$ since $\boldsymbol{u}$ and $\boldsymbol{w}_t$ are both in $\mathbb{B}^d$. In addition, let $\frac{1}{\gamma_0} \triangleq 0$, we notice that

$$\sum_{t=1}^{T} \frac{\gamma_t \|\boldsymbol{v}_t\|^2}{2} = \sum_{t=1}^{T} \gamma_t \left(\frac{1}{\gamma_t^2} - \frac{1}{\gamma_{t-1}^2}\right) \leq \sum_{t=1}^{T} 2\left(\frac{1}{\gamma_t} - \frac{1}{\gamma_{t-1}}\right) = \frac{2}{\gamma_T}. \qquad (11)$$

Combining (10) and (11) and using $\gamma_T = \sqrt{\frac{2}{\sum_{t=1}^{T} \|\boldsymbol{v}_t\|^2}}$, we have

$$\left\langle \sum_{t=1}^{T} \boldsymbol{v}_t, -\boldsymbol{u} \right\rangle \leq 2\sqrt{2\sum_{t=1}^{T} \|\boldsymbol{v}_t\|^2 - \sum_{t=1}^{T} \langle \boldsymbol{v}_t, \boldsymbol{w}_t \rangle}, \forall \boldsymbol{u} \in \mathbb{B}^d, T \in \mathbb{N}.$$

Finally, we use $\max_{\boldsymbol{u} \in \mathbb{B}^d} \left\langle \sum_{t=1}^{T} \boldsymbol{v}_t, -\boldsymbol{u} \right\rangle = \left\| \sum_{t=1}^{T} \boldsymbol{v}_t \right\|$ to obtain

$$\left\| \sum_{t=1}^{T} \boldsymbol{v}_t \right\| \leq 2\sqrt{2\sum_{t=1}^{T} \|\boldsymbol{v}_t\|^2 - \sum_{t=1}^{T} \langle \boldsymbol{v}_t, \boldsymbol{w}_t \rangle}, \forall T \in \mathbb{N}.$$

$\square$

Now we are ready to prove Lemma 4.3.

*Proof of Lemma 4.3.* By Lemma C.1, there exists a sequence of random vectors $\boldsymbol{w}_t \in \mathbb{R}^d$ such that $\boldsymbol{w}_t \in \mathcal{F}_{t-1}$ satisfying

$$\left\| \sum_{t=1}^T \boldsymbol{v}_t \right\| \leq 2\sqrt{2 \sum_{t=1}^T \|\boldsymbol{v}_t\|^2} - \sum_{t=1}^T \langle \boldsymbol{v}_t, \boldsymbol{w}_t \rangle, \forall T \in \mathbb{N}.$$

We take expectations on both sides to get

$$\mathbb{E}\left[ \left\| \sum_{t=1}^T \boldsymbol{v}_t \right\| \right] \leq 2\sqrt{2}\mathbb{E}\left[ \sqrt{\sum_{t=1}^T \|\boldsymbol{v}_t\|^2} \right] - \sum_{t=1}^T \mathbb{E}\left[ \langle \boldsymbol{v}_t, \boldsymbol{w}_t \rangle \right].$$

By noticing that

$$\mathbb{E}\left[ \langle \boldsymbol{v}_t, \boldsymbol{w}_t \rangle \right] = \mathbb{E}\left[ \mathbb{E}\left[ \langle \boldsymbol{v}_t, \boldsymbol{w}_t \rangle \mid \mathcal{F}_{t-1} \right] \right] = \mathbb{E}\left[ \langle \mathbb{E}\left[ \boldsymbol{v}_t \mid \mathcal{F}_{t-1} \right], \boldsymbol{w}_t \rangle \right] = 0,$$

we find

$$\mathbb{E}\left[ \left\| \sum_{t=1}^T \boldsymbol{v}_t \right\| \right] \leq 2\sqrt{2}\mathbb{E}\left[ \sqrt{\sum_{t=1}^T \|\boldsymbol{v}_t\|^2} \right].$$

Finally, by observing $\sqrt{\sum_{t=1}^T \|\boldsymbol{v}_t\|^2} \leq \left( \sum_{t=1}^T \|\boldsymbol{v}_t\|^{\mathfrak{p}} \right)^{\frac{1}{\mathfrak{p}}}, \forall \mathfrak{p} \in [1,2]$, the proof is hence complete.

$\square$

Lastly, we briefly talk about the difference between our Lemma 4.3 and Lemma 4 in Kornilov et al. (2023), the latter of which essentially states an inequality $\mathbb{E}\left[ \left\| \sum_{t=1}^T \boldsymbol{v}_t \right\|^{\mathfrak{p}} \right] \leq 2T\sigma^{\mathfrak{p}}$ under the same setting as in our Lemma 4.3 and an additional condition $\mathbb{E}\left[ \|\boldsymbol{v}_t\|^{\mathfrak{p}} \right] \leq \sigma, \forall t \in [T]$. Though it is possible to employ their inequality to prove convergence under the traditional heavy-tailed noises assumption (i.e., $\sigma_1 = 0$ in Assumption 2.4), whether it is applicable under our new Assumption 2.4 remains unclear since dealing with the new term in the generalized heavy-tailed noises assumption requires a finer conditional expectation argument as shown in the proof of Lemma D.5. Hence, we believe that our new Lemma 4.3 is more general.

# D FULL THEOREMS AND OTHER MISSING PROOFS

## D.1 UPPER BOUNDS

**Theorem D.1.** *(Full version of Theorem 3.2) Under Assumptions 2.1, 2.2, 2.3 and 2.4, let $\Delta_1 \triangleq F(\boldsymbol{x}_1) - F_*$, then for any $T \in \mathbb{N}$, by taking*

$$\beta_t \equiv \beta = 1 - \min\left\{ 1, \max\left\{ \left( \frac{\Delta_1 L_1 B^{\frac{\mathfrak{p}-1}{\mathfrak{p}}} + \sigma_0 + \sigma_1 \|\nabla F(\boldsymbol{x}_1)\|}{\sigma_0 T} \right)^{\frac{\mathfrak{p}}{2\mathfrak{p}-1}}, \left( \frac{\Delta_1 L_0 B^{\frac{2(\mathfrak{p}-1)}{\mathfrak{p}}}}{\sigma_0^2 T} \right)^{\frac{\mathfrak{p}}{3\mathfrak{p}-2}} \right\} \right\},$$

$$\eta_t \equiv \eta = \min\left\{ \sqrt{\frac{(1-\beta)\Delta_1}{L_0 T}}, \frac{1-\beta}{8L_1} \right\}, \quad B = \max\left\{ \left\lceil (16\sqrt{2}\sigma_1)^{\frac{\mathfrak{p}}{\mathfrak{p}-1}} \right\rceil, 1 \right\},$$

*Algorithm 1 guarantees*

$$\frac{1}{T}\sum_{t=1}^T \mathbb{E}\left[ \|\nabla F(\boldsymbol{x}_t)\| \right] = \mathcal{O}\left( \frac{\Delta_1 L_1 + (\sigma_0 + \sigma_1 \|\nabla F(\boldsymbol{x}_1)\|)/B^{\frac{\mathfrak{p}-1}{\mathfrak{p}}}}{T} + \sqrt{\frac{\Delta_1 L_0}{T}} \right.$$

$$\left. + \frac{(\Delta_1 L_1 + (\sigma_0 + \sigma_1 \|\nabla F(\boldsymbol{x}_1)\|)/B^{\frac{\mathfrak{p}-1}{\mathfrak{p}}})^{\frac{\mathfrak{p}-1}{2\mathfrak{p}-1}} \sigma_0^{\frac{\mathfrak{p}}{2\mathfrak{p}-1}}}{(BT)^{\frac{\mathfrak{p}-1}{2\mathfrak{p}-1}}} + \frac{(\Delta_1 L_0)^{\frac{\mathfrak{p}-1}{3\mathfrak{p}-2}} \sigma_0^{\frac{\mathfrak{p}}{3\mathfrak{p}-2}}}{(BT)^{\frac{\mathfrak{p}-1}{3\mathfrak{p}-2}}} \right).$$

*Proof.* Because $\eta_t \equiv \eta \le \frac{1-\beta}{8L_1} \le \frac{1}{L_1}, \forall t \in [T]$, we can safely invoke Lemma D.4 with $\eta_t \equiv \eta, \forall t \in [T]$ to obtain

$$\sum_{t=1}^{T} \eta \mathbb{E}\left[\|\nabla F(\boldsymbol{x}_t)\|\right] \le \Delta_1 + \frac{\eta^2 L_0 T}{2} + \sum_{t=1}^{T} 2\eta \mathbb{E}\left[\|\boldsymbol{\epsilon}_t\|\right] + \sum_{t=1}^{T} \frac{\eta^2 L_1}{2} \mathbb{E}\left[\|\nabla F(\boldsymbol{x}_t)\|\right]. \tag{12}$$

Next, by Lemma D.5 with $\eta_t \equiv \eta, \beta_t \equiv \beta, \forall t \in [T]$, we have for any $t \in [T]$,

$$\begin{aligned}
\mathbb{E}\left[\|\boldsymbol{\epsilon}_t\|\right] \le& \frac{2\sqrt{2}}{B^{\frac{\mathfrak{p}-1}{\mathfrak{p}}}} \left[\beta^t (\sigma_0 + \sigma_1 \|\nabla F(\boldsymbol{x}_1)\|) + (1-\beta) \left(\sum_{s=1}^{t} \beta^{\mathfrak{p}(t-s)}\right)^{\frac{1}{\mathfrak{p}}} \sigma_0\right] \\
&+ \sum_{s=2}^{t} \beta^{t-s+1}(L_0 + L_1 \mathbb{E}\left[\|\nabla F(\boldsymbol{x}_{s-1})\|\right])\eta + \frac{2\sqrt{2}}{B^{\frac{\mathfrak{p}-1}{\mathfrak{p}}}} \sum_{s=1}^{t} (1-\beta)\beta^{t-s}\sigma_1 \mathbb{E}\left[\|\nabla F(\boldsymbol{x}_s)\|\right] \\
\le& \frac{2\sqrt{2}}{B^{\frac{\mathfrak{p}-1}{\mathfrak{p}}}} \left[\beta^t (\sigma_0 + \sigma_1 \|\nabla F(\boldsymbol{x}_1)\|) + \frac{1-\beta}{(1-\beta^{\mathfrak{p}})^{\frac{1}{\mathfrak{p}}}} \sigma_0\right] + \frac{\beta\eta L_0}{1-\beta} \\
&+ \sum_{s=2}^{t} \beta^{t-s+1}\eta L_1 \mathbb{E}\left[\|\nabla F(\boldsymbol{x}_{s-1})\|\right] + \frac{2\sqrt{2}}{B^{\frac{\mathfrak{p}-1}{\mathfrak{p}}}} \sum_{s=1}^{t} (1-\beta)\beta^{t-s}\sigma_1 \mathbb{E}\left[\|\nabla F(\boldsymbol{x}_s)\|\right] \\
\le& \frac{2\sqrt{2}}{B^{\frac{\mathfrak{p}-1}{\mathfrak{p}}}} \left[\beta^t (\sigma_0 + \sigma_1 \|\nabla F(\boldsymbol{x}_1)\|) + (1-\beta)^{\frac{\mathfrak{p}-1}{\mathfrak{p}}} \sigma_0\right] + \frac{\beta\eta L_0}{1-\beta} \\
&+ \sum_{s=2}^{t} \beta^{t-s+1}\eta L_1 \mathbb{E}\left[\|\nabla F(\boldsymbol{x}_{s-1})\|\right] + \frac{2\sqrt{2}}{B^{\frac{\mathfrak{p}-1}{\mathfrak{p}}}} \sum_{s=1}^{t} (1-\beta)\beta^{t-s}\sigma_1 \mathbb{E}\left[\|\nabla F(\boldsymbol{x}_s)\|\right],
\end{aligned}$$

where we use $\beta^{\mathfrak{p}} \le \beta$ when $\mathfrak{p} \ge 1$ and $\beta \le 1$ in the last step. As such, we know

$$\begin{aligned}
\sum_{t=1}^{T} 2\eta \mathbb{E}\left[\|\boldsymbol{\epsilon}_t\|\right] \le& \frac{4\sqrt{2}\eta}{B^{\frac{\mathfrak{p}-1}{\mathfrak{p}}}} \sum_{t=1}^{T} \left[\beta^t (\sigma_0 + \sigma_1 \|\nabla F(\boldsymbol{x}_1)\|) + (1-\beta)^{\frac{\mathfrak{p}-1}{\mathfrak{p}}} \sigma_0\right] + \sum_{t=1}^{T} \frac{2\beta\eta^2 L_0}{1-\beta} \\
&+ \sum_{t=1}^{T} \sum_{s=2}^{t} 2\beta^{t-s+1}\eta^2 L_1 \mathbb{E}\left[\|\nabla F(\boldsymbol{x}_{s-1})\|\right] + \frac{4\sqrt{2}\eta}{B^{\frac{\mathfrak{p}-1}{\mathfrak{p}}}} \sum_{t=1}^{T} \sum_{s=1}^{t} (1-\beta)\beta^{t-s}\sigma_1 \mathbb{E}\left[\|\nabla F(\boldsymbol{x}_s)\|\right] \\
\le& \frac{4\sqrt{2}\eta}{B^{\frac{\mathfrak{p}-1}{\mathfrak{p}}}} \left[\frac{\beta (\sigma_0 + \sigma_1 \|\nabla F(\boldsymbol{x}_1)\|)}{1-\beta} + (1-\beta)^{\frac{\mathfrak{p}-1}{\mathfrak{p}}} T\sigma_0\right] + \frac{2\beta\eta^2 L_0 T}{1-\beta} \\
&+ \sum_{t=1}^{T} \left(\frac{2\beta\eta^2 L_1}{1-\beta} + \frac{4\sqrt{2}\eta\sigma_1}{B^{\frac{\mathfrak{p}-1}{\mathfrak{p}}}}\right) \mathbb{E}\left[\|\nabla F(\boldsymbol{x}_t)\|\right], \tag{13}
\end{aligned}$$

where in the last step we use $\sum_{t=1}^{T} \sum_{s=i}^{t} \cdot = \sum_{s=i}^{T} \sum_{t=s}^{T} \cdot \le \sum_{s=i}^{T} \sum_{t=s}^{\infty} \cdot$ when the summands are non-negative for $i \in \{1, 2\}$.

Now plugging (13) into (12) to get

$$\sum_{t=1}^{T} \eta \mathbb{E}\left[\|\nabla F(\boldsymbol{x}_t)\|\right]$$

$$\leq \Delta_1 + \frac{(1+3\beta)\eta^2 L_0 T}{2(1-\beta)} + \frac{4\sqrt{2}\eta}{B^{\frac{p-1}{p}}}\left[\frac{\beta\left(\sigma_0 + \sigma_1 \|\nabla F(\boldsymbol{x}_1)\|\right)}{1-\beta} + (1-\beta)^{\frac{p-1}{p}} T\sigma_0\right]$$

$$+ \sum_{t=1}^{T}\left(\frac{(1+3\beta)\eta^2 L_1}{2(1-\beta)} + \frac{4\sqrt{2}\eta\sigma_1}{B^{\frac{p-1}{p}}}\right)\mathbb{E}\left[\|\nabla F(\boldsymbol{x}_t)\|\right]$$

$$\overset{(a)}{\leq} \Delta_1 + \frac{2\eta^2 L_0 T}{1-\beta} + \frac{4\sqrt{2}\eta}{B^{\frac{p-1}{p}}}\left[\frac{\sigma_0 + \sigma_1 \|\nabla F(\boldsymbol{x}_1)\|}{1-\beta} + (1-\beta)^{\frac{p-1}{p}} T\sigma_0\right]$$

$$+ \sum_{t=1}^{T}\left(\frac{2\eta^2 L_1}{1-\beta} + \frac{4\sqrt{2}\eta\sigma_1}{B^{\frac{p-1}{p}}}\right)\mathbb{E}\left[\|\nabla F(\boldsymbol{x}_t)\|\right]$$

$$\overset{(b)}{\leq} \Delta_1 + \frac{2\eta^2 L_0 T}{1-\beta} + \frac{4\sqrt{2}\eta}{B^{\frac{p-1}{p}}}\left[\frac{\sigma_0 + \sigma_1 \|\nabla F(\boldsymbol{x}_1)\|}{1-\beta} + (1-\beta)^{\frac{p-1}{p}} T\sigma_0\right] + \sum_{t=1}^{T}\frac{\eta}{2}\mathbb{E}\left[\|\nabla F(\boldsymbol{x}_t)\|\right],$$

$$\tag{14}$$

where we use $\beta \leq 1$ in $(a)$ and $\eta \leq \frac{1-\beta}{8L_1}, B \geq (16\sqrt{2}\sigma_1)^{\frac{p}{p-1}}$ in $(b)$. We observe that (14) implies

$$\sum_{t=1}^{T}\mathbb{E}\left[\|\nabla F(\boldsymbol{x}_t)\|\right]$$

$$\leq \frac{2\Delta_1}{\eta} + \frac{4\eta L_0 T}{1-\beta} + \frac{8\sqrt{2}}{B^{\frac{p-1}{p}}}\left[\frac{\sigma_0 + \sigma_1 \|\nabla F(\boldsymbol{x}_1)\|}{1-\beta} + (1-\beta)^{\frac{p-1}{p}} T\sigma_0\right] \tag{15}$$

$$\overset{(c)}{=} \mathcal{O}\left(\frac{\Delta_1 L_1 + (\sigma_0 + \sigma_1 \|\nabla F(\boldsymbol{x}_1)\|)/B^{\frac{p-1}{p}}}{1-\beta} + \sqrt{\frac{\Delta L_0 T}{1-\beta}} + \frac{(1-\beta)^{\frac{p-1}{p}}\sigma_0 T}{B^{\frac{p-1}{p}}}\right)$$

$$\overset{(d)}{=} \mathcal{O}\left(\Delta_1 L_1 + (\sigma_0 + \sigma_1 \|\nabla F(\boldsymbol{x}_1)\|)/B^{\frac{p-1}{p}} + \sqrt{\Delta L_0 T}\right.$$

$$\left. + \frac{(\Delta_1 L_1 + (\sigma_0 + \sigma_1 \|\nabla F(\boldsymbol{x}_1)\|)/B^{\frac{p-1}{p}})^{\frac{p-1}{2p-1}}(\sigma_0 T)^{\frac{p}{2p-1}}}{B^{\frac{p-1}{2p-1}}} + \frac{(\Delta_1 L_0)^{\frac{p-1}{3p-2}}\sigma_0^{\frac{p}{3p-2}}T^{\frac{2p-1}{3p-2}}}{B^{\frac{p-1}{3p-2}}}\right),$$

$$\tag{16}$$

where we plug in $\eta = \min\left\{\sqrt{\frac{(1-\beta)\Delta_1}{L_0 T}}, \frac{1-\beta}{8L_1}\right\}$ in $(c)$ and use the following estimation in $(d)$:

Let $U \triangleq \left(\frac{\Delta_1 L_1 B^{\frac{p-1}{p}} + \sigma_0 + \sigma_1 \|\nabla F(\boldsymbol{x}_1)\|}{\sigma_0 T}\right)^{\frac{p}{2p-1}}$ and $V \triangleq \left(\frac{\Delta_1 L_0 B^{\frac{2(p-1)}{p}}}{\sigma_0^2 T}\right)^{\frac{p}{3p-2}}$, we know $1-\beta = \min\{1, \max\{U, V\}\}$. Then there is

$$\frac{C_1}{1-\beta} + \sqrt{\frac{C_2}{1-\beta}} + C_3(1-\beta)^{\frac{p-1}{p}}$$

$$\leq C_1\left(1 + \frac{1}{\max\{U, V\}}\right) + \sqrt{C_2\left(1 + \frac{1}{\max\{U, V\}}\right)} + C_3\left(\max\{U, V\}\right)^{\frac{p-1}{p}}$$

$$\leq C_1\left(1 + \frac{1}{U}\right) + \sqrt{C_2\left(1 + \frac{1}{V}\right)} + C_3\left(U^{\frac{p-1}{p}} + V^{\frac{p-1}{p}}\right)$$

$$\leq C_1 + \sqrt{C_2} + \frac{C_1}{U} + \sqrt{\frac{C_2}{V}} + C_3\left(U^{\frac{p-1}{p}} + V^{\frac{p-1}{p}}\right),$$

where

$$C_1 \triangleq \Delta_1 L_1 + (\sigma_0 + \sigma_1 \|\nabla F(\boldsymbol{x}_1)\|)/B^{\frac{p-1}{p}}, \quad C_2 \triangleq \Delta L_0 T, \quad C_3 \triangleq \frac{\sigma_0 T}{B^{\frac{p-1}{p}}}.$$

Finally, divide both sides of (16) by $T$ to obtain the desired result. □

**Theorem D.2.** *(Full version of Theorem 3.6) Under Assumptions 2.1, 2.2, 2.3 and 2.4, let* $\Delta_1 \triangleq F(\boldsymbol{x}_1) - F_*$, *then for any* $T \in \mathbb{N}$, *by taking*

$$
\beta_t \equiv \beta = 1 - \frac{1}{T^{\frac{1}{2}}}, \quad \eta_t \equiv \eta = \min\left\{\frac{1}{T^{\frac{3}{4}}}, \frac{1}{8L_1 T^{\frac{1}{2}}}\right\}, \quad B = \max\left\{\left\lceil (16\sqrt{2}\sigma_1)^{\frac{\mathfrak{p}}{\mathfrak{p}-1}} \right\rceil, 1\right\},
$$

*Algorithm 1 guarantees*

$$
\frac{1}{T}\sum_{t=1}^{T}\mathbb{E}\left[\|\nabla F(\boldsymbol{x}_t)\|\right] = \mathcal{O}\left(\frac{\Delta_1 L_1 + (\sigma_0 + \sigma_1\|\nabla F(\boldsymbol{x}_1)\|)/B^{\frac{\mathfrak{p}-1}{\mathfrak{p}}}}{\sqrt{T}} + \frac{\Delta_1 + L_0}{T^{\frac{1}{4}}} + \frac{\sigma_0}{(BT)^{\frac{\mathfrak{p}-1}{\mathfrak{p}}}}\right).
$$

*In particular, if* $\sigma_1 \leq \frac{1}{16\sqrt{2}}$, *we can always set* $B = 1$ *to get rid of the tail index* $\mathfrak{p}$.

*Remark D.3.* We note that it is possible to improve the threshold $\frac{1}{16\sqrt{2}}$ to a slightly bigger constant, but this still cannot help us to remove the requirement of needing $\mathfrak{p}$ when $\sigma_1$ becomes larger. Thus, we do not put further effort into optimizing this constant but keep it as it is.

*Proof.* Note that (15) still holds under current parameter choices since $\eta = \min\left\{\frac{1}{T^{\frac{3}{4}}}, \frac{1}{8L_1 T^{\frac{1}{2}}}\right\} = \min\left\{\sqrt{\frac{1-\beta}{T}}, \frac{1-\beta}{8L_1}\right\}$. Hence, we have

$$
\sum_{t=1}^{T}\mathbb{E}\left[\|\nabla F(\boldsymbol{x}_t)\|\right]
$$

$$
\leq \mathcal{O}\left(\frac{\Delta_1}{\eta} + \frac{\eta L_0 T}{1-\beta} + \frac{1}{B^{\frac{\mathfrak{p}-1}{\mathfrak{p}}}}\left[\frac{\sigma_0 + \sigma_1\|\nabla F(\boldsymbol{x}_1)\|}{1-\beta} + (1-\beta)^{\frac{\mathfrak{p}-1}{\mathfrak{p}}}T\sigma_0\right]\right)
$$

$$
\overset{(a)}{=}\mathcal{O}\left(\frac{\Delta_1 L_1 + (\sigma_0 + \sigma_1\|\nabla F(\boldsymbol{x}_1)\|)/B^{\frac{\mathfrak{p}-1}{\mathfrak{p}}}}{1-\beta} + (\Delta_1 + L_0)\sqrt{\frac{T}{1-\beta}} + \frac{(1-\beta)^{\frac{\mathfrak{p}-1}{\mathfrak{p}}}\sigma_0 T}{B^{\frac{\mathfrak{p}-1}{\mathfrak{p}}}}\right)
$$

$$
\overset{(b)}{=}\mathcal{O}\left(\left[\Delta_1 L_1 + (\sigma_0 + \sigma_1\|\nabla F(\boldsymbol{x}_1)\|)/B^{\frac{\mathfrak{p}-1}{\mathfrak{p}}}\right]\sqrt{T} + (\Delta_1 + L_0)T^{\frac{3}{4}} + \frac{\sigma_0 T^{\frac{\mathfrak{p}+1}{2\mathfrak{p}}}}{B^{\frac{\mathfrak{p}-1}{\mathfrak{p}}}}\right),
$$

where we use $\eta = \min\left\{\frac{1}{T^{\frac{3}{4}}}, \frac{1}{8L_1 T^{\frac{1}{2}}}\right\} = \min\left\{\sqrt{\frac{1-\beta}{T}}, \frac{1-\beta}{8L_1}\right\}$ in $(a)$ and $1-\beta = \frac{1}{T^{\frac{1}{2}}}$ in $(b)$. We divide both sides by $T$ to obtain the desired result. □

## D.2 LOWER BOUND

In this subsection, we will prove the lower bound, Theorem 3.3. The proof is a simple variation of Zhang et al. (2020c), which itself is based on Carmon et al. (2020); Arjevani et al. (2023).

*Proof of Theorem 3.3.* For any $\boldsymbol{x} \in \mathbb{R}^d$ and $\alpha \in [0,1]$, we denote by $\text{prog}_\alpha(\boldsymbol{x})$ the highest index whose entry is $\alpha$-far from 0, i.e.,

$$
\text{prog}_\alpha(\boldsymbol{x}) \triangleq \max\left\{i \in [d] : |\boldsymbol{x}[i]| > \alpha\right\} \text{ where } \max\emptyset \triangleq 0.
$$

Now given $d \in \mathbb{N}$, we introduce the following underlying function originally proposed by Carmon et al. (2020).

$$
f_d(\boldsymbol{x}) \triangleq -\Psi(1)\Phi(\boldsymbol{x}[1]) + \sum_{i=2}^{d}\Psi(-\boldsymbol{x}[i-1])\Phi(-\boldsymbol{x}[i]) - \Psi(\boldsymbol{x}[i-1])\Phi(\boldsymbol{x}[i]),
$$

where

$$
\Psi(t) \triangleq \begin{cases} 0 & t \leq \frac{1}{2} \\ \exp(1 - (2t-1)^{-2}) & t > \frac{1}{2} \end{cases}, \quad \Phi(t) \triangleq \sqrt{e}\int_{-\infty}^{t}\exp(-\tau^2/2)\mathrm{d}\tau.
$$

By Lemma 2 in Arjevani et al. (2023), $f_d$ admits the following properties:

1. $f_d(\mathbf{0}) - f_{d,*} \leq \delta d$, where $f_{d,*} \triangleq \inf_{\boldsymbol{x} \in \mathbb{R}^d} f_d(\boldsymbol{x})$ and $\delta = 12$.

2. $f_d$ is $\ell$-smooth, where $\ell = 152$.

3. For all $\boldsymbol{x} \in \mathbb{R}^d$, $\|\nabla f_d(\boldsymbol{x})\|_\infty \leq \gamma$, where $\gamma = 23$.

4. For all $\boldsymbol{x} \in \mathbb{R}^d$, $\mathrm{prog}_0(\nabla f_d(\boldsymbol{x})) \leq \mathrm{prog}_{\frac{1}{2}}(\boldsymbol{x}) + 1$.

5. For all $\boldsymbol{x} \in \mathbb{R}^d$ and $i \triangleq \mathrm{prog}_{\frac{1}{2}}(\boldsymbol{x})$, $\nabla f_d(\boldsymbol{x}) = \nabla f_d(\boldsymbol{x}_{\leq 1+i})$ and $[\nabla f_d(\boldsymbol{x})]_{\leq i} = [\nabla f_d(\boldsymbol{x}_{\leq i})]_{\leq i}$, where $\boldsymbol{y}_{\leq i}[j] \triangleq \boldsymbol{y}[j]\mathbb{1}[j \leq i]$.

6. For all $\boldsymbol{x} \in \mathbb{R}^d$, if $\mathrm{prog}_1(\boldsymbol{x}) < d$, then $\|\nabla f_d(\boldsymbol{x})\| > 1$.

Now we consider the following stochastic oracle introduced by Arjevani et al. (2023),

$$\boldsymbol{h}_d(\boldsymbol{x}, z)[i] \triangleq \nabla f_d(\boldsymbol{x})[i]\left(1 + \mathbb{1}\left[i > \mathrm{prog}_{\frac{1}{4}}(\boldsymbol{x})\right]\left(\frac{z}{q} - 1\right)\right), \forall i \in [d],$$

where $z = \mathrm{Bernoulli}(q)$ and $q \in [0, 1]$ will be specified later. As one can check, $\mathbb{E}_z[\boldsymbol{h}_d(\boldsymbol{x}, z)] = \nabla f_d(\boldsymbol{x}), \forall \boldsymbol{x} \in \mathbb{R}^d$. Moreover, by Lemma 3 in Arjevani et al. (2023), we know $\boldsymbol{h}_d$ is a probability-$q$ zero-chain (see Definition 2 in Arjevani et al. (2023) for what it is) and satisfies almost surely

$$\|\boldsymbol{h}_d(\boldsymbol{x}, z) - \nabla f_d(\boldsymbol{x})\| \leq \gamma\left|\frac{z}{q} - 1\right|, \forall \boldsymbol{x} \in \mathbb{R}^d. \tag{17}$$

Next, given $\mathfrak{p} \in (1, 2]$, $\Delta_1, L_0, \sigma_0 > 0$, and small enough $\varepsilon$, we define $d \triangleq \left\lfloor \frac{\Delta_1 L_0}{4\delta\ell\varepsilon^2} \right\rfloor$ and

$$F_d(\boldsymbol{x}) \triangleq \frac{L_0\lambda^2}{\ell} f_d\left(\frac{\boldsymbol{x}}{\lambda}\right), \text{ where } \lambda \triangleq \frac{2\ell\varepsilon}{L_0}.$$

Moreover, we let

$$\boldsymbol{g}_d(\boldsymbol{x}, z) \triangleq \frac{L_0\lambda}{\ell} \boldsymbol{h}_d\left(\frac{\boldsymbol{x}}{\lambda}, z\right), \text{ and } q \triangleq \left(\frac{4\gamma\varepsilon}{\sigma_0}\right)^{\frac{\mathfrak{p}}{\mathfrak{p}-1}}.$$

$q \leq 1$ can be true since we assume that $\varepsilon$ is small enough.

Note that $F_d$ is lower bounded since $f_d$ is lower bounded and thus satisfies Assumption 2.1. In addition, we have $\nabla F_d(\boldsymbol{x}) = \frac{L_0\lambda}{\ell}\nabla f_d\left(\frac{\boldsymbol{x}}{\lambda}\right)$, which implies $F_d$ is $L_0$-smooth because $f_d$ is $\ell$-smooth. So $F_d$ also satisfies Assumption 2.2 with $L_1 = 0$. Moreover, we can find

$$F_d(\mathbf{0}) - \inf_{\boldsymbol{x} \in \mathbb{R}^d} F_d(\boldsymbol{x}) = \frac{L_0\lambda^2}{\ell}\left(f_d\left(\frac{\boldsymbol{x}}{\lambda}\right) - f_{d,*}\right) \leq \frac{L_0\lambda^2}{\ell} \cdot \delta d \leq \frac{L_0\lambda^2}{\ell} \cdot \delta \cdot \frac{\Delta_1 L_0}{4\delta\ell\varepsilon^2} = \Delta_1.$$

Now let us verify

$$\mathbb{E}_z[\boldsymbol{g}_d(\boldsymbol{x}, z)] = \mathbb{E}_z\left[\frac{L_0\lambda}{\ell}\boldsymbol{h}_d\left(\frac{\boldsymbol{x}}{\lambda}, z\right)\right] = \frac{L_0\lambda}{\ell}\nabla f_d\left(\frac{\boldsymbol{x}}{\lambda}\right) = \nabla F_d(\boldsymbol{x}).$$

Hence, $\boldsymbol{g}_d(\boldsymbol{x}, z)$ satisfies Assumption 2.3. Lastly, we know

$$\mathbb{E}_z\left[\|\boldsymbol{g}_d(\boldsymbol{x}, z) - \nabla F_d(\boldsymbol{x})\|^{\mathfrak{p}}\right] = \left(\frac{L_0\lambda}{\ell}\right)^{\mathfrak{p}} \mathbb{E}_z\left[\left\|\boldsymbol{h}_d\left(\frac{\boldsymbol{x}}{\lambda}, z\right) - \nabla f_d\left(\frac{\boldsymbol{x}}{\lambda}\right)\right\|^{\mathfrak{p}}\right] \overset{(17)}{\leq} \left(\frac{L_0\lambda\gamma}{\ell}\right)^{\mathfrak{p}} \mathbb{E}\left[\left|\frac{z}{q} - 1\right|^{\mathfrak{p}}\right]$$

$$= \left(\frac{L_0\lambda\gamma}{\ell}\right)^{\mathfrak{p}}\left(1 - q + \frac{(1-q)^{\mathfrak{p}}}{q^{\mathfrak{p}-1}}\right) = (2\gamma\varepsilon)^{\mathfrak{p}}(1-q)\frac{q^{\mathfrak{p}-1} + (1-q)^{\mathfrak{p}-1}}{q^{\mathfrak{p}-1}}$$

$$\leq \frac{(4\gamma\varepsilon)^{\mathfrak{p}}}{q^{\mathfrak{p}-1}} = \sigma_0^{\mathfrak{p}}.$$

Thus, $\boldsymbol{g}_d(\boldsymbol{x}, z)$ satisfies Assumption 2.4 with $\sigma_1 = 0$.

Finally, by Lemma 1 in Arjevani et al. (2023), for any zero-respecting algorithm starting from $\boldsymbol{x}_1 = \mathbf{0}$, with probability at least $\frac{1}{2}$, $\mathrm{prog}_0(\boldsymbol{x}_t) < d, \forall t \leq \frac{d-1}{2q}$. By noticing that $\mathrm{prog}_1\left(\frac{\boldsymbol{x}_t}{\lambda}\right) \leq$

$\operatorname{prog}_0\left(\frac{\boldsymbol{x}_t}{\lambda}\right) = \operatorname{prog}_0(\boldsymbol{x}_t) < d$, we then have with probability at least $\frac{1}{2}$, $\left\|\nabla f_d\left(\frac{\boldsymbol{x}_t}{\lambda}\right)\right\| > 1, \forall t \le \frac{d-1}{2q}$, which implies

$$\mathbb{E}\left[\|\nabla F_d(\boldsymbol{x}_t)\|\right] = \frac{L_0\lambda}{\ell}\mathbb{E}\left[\left\|\nabla f_d\left(\frac{\boldsymbol{x}_t}{\lambda}\right)\right\|\right] > \frac{L_0\lambda}{2\ell} = \varepsilon, \forall t \le \frac{d-1}{2q}.$$

Thus, to output an $\varepsilon$-stationary point, the algorithm need at least the following number of iterations

$$\frac{d-1}{2q} = \frac{1}{2}\left(\left\lfloor\frac{\Delta_1 L_0}{4\delta\ell\varepsilon^2}\right\rfloor - 1\right)\cdot\left(\frac{\sigma_0}{4\gamma\varepsilon}\right)^{\frac{\mathsf{p}}{\mathsf{p}-1}} = \Omega\left(\Delta_1 L_0\sigma_0^{\frac{\mathsf{p}}{\mathsf{p}-1}}\varepsilon^{-\frac{3\mathsf{p}-2}{\mathsf{p}-1}}\right).$$

$\square$

## D.3 HELPFUL LEMMAS

We first recall the notations as follows

$$\boldsymbol{\epsilon}_t \triangleq \begin{cases} \boldsymbol{m}_t - \nabla F(\boldsymbol{x}_t) & t \in [T] \\ \boldsymbol{m}_0 - \nabla F(\boldsymbol{x}_1) & t = 0 \end{cases}, \tag{18}$$

$$\boldsymbol{D}_t \triangleq \mathbb{1}_{t\ge2}\left(\nabla F(\boldsymbol{x}_{t-1}) - \nabla F(\boldsymbol{x}_t)\right), \forall t \in [T], \tag{19}$$

$$\boldsymbol{\xi}_t^i \triangleq \boldsymbol{g}_t^i - \nabla F(\boldsymbol{x}_t), \forall i \in [B], \forall t \in [T], \tag{20}$$

$$\boldsymbol{\xi}_t \triangleq \boldsymbol{g}_t - \nabla F(\boldsymbol{x}_t) = \frac{1}{B}\sum_{i=1}^B \boldsymbol{\xi}_t^i, \forall t \in [T]. \tag{21}$$

Now we are ready to start the proof.

**Lemma D.4.** *(Full version of Lemma 4.1) Under Assumptions 2.1 and 2.2, let $\Delta_1 \triangleq F(\boldsymbol{x}_1) - F_*$, if $\eta_t \le \frac{1}{L_1}, \forall t \in [T]$, then Algorithm 1 guarantees*

$$\sum_{t=1}^T \eta_t\mathbb{E}\left[\|\nabla F(\boldsymbol{x}_t)\|\right] \le \Delta_1 + \sum_{t=1}^T 2\eta_t\mathbb{E}\left[\|\boldsymbol{\epsilon}_t\|\right] + \sum_{t=1}^T \frac{L_0 + L_1\mathbb{E}\left[\|\nabla F(\boldsymbol{x}_t)\|\right]}{2}\eta_t^2.$$

*Proof.* Note that $\|\boldsymbol{x}_{t+1} - \boldsymbol{x}_t\| = \left\|\eta_t\frac{\boldsymbol{m}_t}{\|\boldsymbol{m}_t\|}\right\| \le \eta_t \le \frac{1}{L_1}$ now, we then invoke Lemma (2.5) to obtain for any $t \in [T]$,

$$F(\boldsymbol{x}_{t+1}) \le F(\boldsymbol{x}_t) + \langle\nabla F(\boldsymbol{x}_t), \boldsymbol{x}_{t+1} - \boldsymbol{x}_t\rangle + \frac{L_0 + L_1\|\nabla F(\boldsymbol{x}_t)\|}{2}\|\boldsymbol{x}_{t+1} - \boldsymbol{x}_t\|^2$$

$$= F(\boldsymbol{x}_t) - \eta_t\langle\nabla F(\boldsymbol{x}_t), \boldsymbol{m}_t/\|\boldsymbol{m}_t\|\rangle + \frac{L_0 + L_1\|\nabla F(\boldsymbol{x}_t)\|}{2}\eta_t^2$$

$$\overset{(18)}{=} F(\boldsymbol{x}_t) - \eta_t\|\boldsymbol{m}_t\| + \eta_t\langle\boldsymbol{\epsilon}_t, \boldsymbol{m}_t/\|\boldsymbol{m}_t\|\rangle + \frac{L_0 + L_1\|\nabla F(\boldsymbol{x}_t)\|}{2}\eta_t^2$$

$$\overset{(a)}{\le} F(\boldsymbol{x}_t) - \eta_t\|\boldsymbol{m}_t\| + \eta_t\|\boldsymbol{\epsilon}_t\| + \frac{L_0 + L_1\|\nabla F(\boldsymbol{x}_t)\|}{2}\eta_t^2$$

$$\overset{(b)}{\le} F(\boldsymbol{x}_t) - \eta_t\|\nabla F(\boldsymbol{x}_t)\| + 2\eta_t\|\boldsymbol{\epsilon}_t\| + \frac{L_0 + L_1\|\nabla F(\boldsymbol{x}_t)\|}{2}\eta_t^2, \tag{22}$$

where $(a)$ is by $\langle\boldsymbol{\epsilon}_t, \boldsymbol{m}_t/\|\boldsymbol{m}_t\|\rangle \le \|\boldsymbol{\epsilon}_t\|\|\boldsymbol{m}_t/\|\boldsymbol{m}_t\|\| = \|\boldsymbol{\epsilon}_t\|$ and $(b)$ is due to $\|\boldsymbol{m}_t\| \overset{(18)}{=} \|\nabla F(\boldsymbol{x}_t) + \boldsymbol{\epsilon}_t\| \ge \|\nabla F(\boldsymbol{x}_t)\| - \|\boldsymbol{\epsilon}_t\|$. Taking expectations on both sides of (22) and summing up from $t = 1$ to $T$, we have

$$\mathbb{E}\left[F(\boldsymbol{x}_{T+1})\right] \le F(\boldsymbol{x}_1) - \sum_{t=1}^T \eta_t\mathbb{E}\left[\|\nabla F(\boldsymbol{x}_t)\|\right] + \sum_{t=1}^T 2\eta_t\mathbb{E}\left[\|\boldsymbol{\epsilon}_t\|\right] + \sum_{t=1}^T \frac{L_0 + L_1\mathbb{E}\left[\|\nabla F(\boldsymbol{x}_t)\|\right]}{2}\eta_t^2.$$

Finally, we rearrange the terms, apply $\mathbb{E}\left[F(\boldsymbol{x}_{T+1})\right] \ge F_*$ due to Assumption 2.1 and use $\Delta_1 = F(\boldsymbol{x}_1) - F_*$ to get the desired result. $\square$

**Lemma D.5.** *(Full version of Lemma 4.5) Under Assumptions 2.2, 2.3 and 2.4, if $\eta_t \leq \frac{1}{L_1}, \forall t \in [T]$, then Algorithm 1 guarantees*

$$
\mathbb{E}\left[\|\boldsymbol{\epsilon}_t\|\right] \leq \frac{2\sqrt{2}}{B^{\frac{\mathfrak{p}-1}{\mathfrak{p}}}} \left[\beta_{1:t}\left(\sigma_0 + \sigma_1 \|\nabla F(\boldsymbol{x}_1)\|\right) + \left(\sum_{s=1}^{t}(1-\beta_s)^{\mathfrak{p}}(\beta_{s+1:t})^{\mathfrak{p}}\right)^{\frac{1}{\mathfrak{p}}}\sigma_0\right]
$$

$$
+ \sum_{s=2}^{t}\beta_{s:t}\left(L_0 + L_1\mathbb{E}\left[\|\nabla F(\boldsymbol{x}_{s-1})\|\right]\right)\eta_{s-1}
$$

$$
+ \frac{2\sqrt{2}}{B^{\frac{\mathfrak{p}-1}{\mathfrak{p}}}}\sum_{s=1}^{t}(1-\beta_s)\beta_{s+1:t}\sigma_1\mathbb{E}\left[\|\nabla F(\boldsymbol{x}_s)\|\right], \forall t \in [T].
$$

*Proof.* Based on Lemma 4.2, we know for any $t \in [T]$,

$$
\|\boldsymbol{\epsilon}_t\| \leq \beta_{1:t}\|\boldsymbol{\epsilon}_0\| + \left\|\sum_{s=1}^{t}\beta_{s:t}\boldsymbol{D}_s\right\| + \left\|\sum_{s=1}^{t}(1-\beta_s)\beta_{s+1:t}\boldsymbol{\xi}_s\right\|
$$

$$
\Rightarrow \mathbb{E}\left[\|\boldsymbol{\epsilon}_t\|\right] \leq \beta_{1:t}\mathbb{E}\left[\|\boldsymbol{\epsilon}_0\|\right] + \mathbb{E}\left[\left\|\sum_{s=1}^{t}\beta_{s:t}\boldsymbol{D}_s\right\|\right] + \mathbb{E}\left[\left\|\sum_{s=1}^{t}(1-\beta_s)\beta_{s+1:t}\boldsymbol{\xi}_s\right\|\right]. \tag{23}
$$

First, by the definition of $\boldsymbol{\epsilon}_0 \stackrel{(18)}{=} \boldsymbol{m}_0 - \nabla F(\boldsymbol{x}_1) = \boldsymbol{g}_1 - \nabla F(\boldsymbol{x}_1) \stackrel{(20)}{=} \frac{1}{B}\sum_{i=1}^{B}\boldsymbol{\xi}_1^i$, we have

$$
\mathbb{E}\left[\|\boldsymbol{\epsilon}_0\|\right] = \frac{1}{B}\mathbb{E}\left[\left\|\sum_{i=1}^{B}\boldsymbol{\xi}_1^i\right\|\right] \stackrel{(a)}{\leq} \frac{2\sqrt{2}}{B}\mathbb{E}\left[\left(\sum_{i=1}^{B}\|\boldsymbol{\xi}_1^i\|^{\mathfrak{p}}\right)^{\frac{1}{\mathfrak{p}}}\right]
$$

$$
\stackrel{(b)}{\leq} \frac{2\sqrt{2}}{B}\left(\sum_{i=1}^{B}\mathbb{E}\left[\|\boldsymbol{\xi}_1^i\|^{\mathfrak{p}}\right]\right)^{\frac{1}{\mathfrak{p}}} \stackrel{\text{Assumption 2.4}}{\leq} \frac{2\sqrt{2}}{B^{\frac{\mathfrak{p}-1}{\mathfrak{p}}}}\left(\sigma_0^{\mathfrak{p}} + \sigma_1^{\mathfrak{p}}\|\nabla F(\boldsymbol{x}_1)\|^{\mathfrak{p}}\right)^{\frac{1}{\mathfrak{p}}}
$$

$$
\stackrel{(c)}{\leq} \frac{2\sqrt{2}}{B^{\frac{\mathfrak{p}-1}{\mathfrak{p}}}}\left(\sigma_0 + \sigma_1\|\nabla F(\boldsymbol{x}_1)\|\right), \tag{24}
$$

where $(a)$ is by applying Lemma 4.3 with $\boldsymbol{v}_i \triangleq \boldsymbol{\xi}_1^i, \forall i \in [B]$, $(b)$ is due to Hölder's inequality, and $(c)$ is because of $(x+y)^{\frac{1}{\mathfrak{p}}} \leq x^{\frac{1}{\mathfrak{p}}} + y^{\frac{1}{\mathfrak{p}}}$ when $\mathfrak{p} \geq 1$.

Next, we know

$$
\left\|\sum_{s=1}^{t}\beta_{s:t}\boldsymbol{D}_s\right\| \leq \sum_{s=1}^{t}\beta_{s:t}\|\boldsymbol{D}_s\| \stackrel{(19)}{=} \sum_{s=1}^{t}\beta_{s:t}\|\nabla F(\boldsymbol{x}_{s-1}) - \nabla F(\boldsymbol{x}_s)\|\mathbb{1}_{s \geq 2}
$$

$$
\stackrel{\text{Assumption 2.2}}{\leq} \sum_{s=1}^{t}\beta_{s:t}\left(L_0 + L_1\|\nabla F(\boldsymbol{x}_{s-1})\|\right)\|\boldsymbol{x}_s - \boldsymbol{x}_{s-1}\|\mathbb{1}_{s \geq 2}
$$

$$
\stackrel{(d)}{\leq} \sum_{s=1}^{t}\beta_{s:t}\left(L_0 + L_1\|\nabla F(\boldsymbol{x}_{s-1})\|\right)\eta_{s-1}\mathbb{1}_{s \geq 2}
$$

$$
= \sum_{s=2}^{t}\beta_{s:t}\left(L_0 + L_1\|\nabla F(\boldsymbol{x}_{s-1})\|\right)\eta_{s-1},
$$

where $(d)$ is by $\|\boldsymbol{x}_s - \boldsymbol{x}_{s-1}\| = \left\|\eta_{s-1}\frac{\boldsymbol{m}_{s-1}}{\|\boldsymbol{m}_{s-1}\|}\right\| \leq \eta_{s-1}$ from the update rule of Algorithm 1. Therefore, we have

$$
\mathbb{E}\left[\left\|\sum_{s=1}^{t}\beta_{s:t}\boldsymbol{D}_s\right\|\right] \leq \sum_{s=2}^{t}\beta_{s:t}\left(L_0 + L_1\mathbb{E}\left[\|\nabla F(\boldsymbol{x}_{s-1})\|\right]\right)\eta_{s-1}. \tag{25}
$$

Moreover, let $\boldsymbol{v}_{(s-1)B+i} \triangleq (1 - \beta_s)\beta_{s+1:t}\boldsymbol{\xi}_s^i, \forall i \in [B], s \in [t]$ and note that this sequence satisfies the requirement of Lemma 4.3, then there is

$$
\mathbb{E}\left[\left\|\sum_{s=1}^{t}(1 - \beta_s)\beta_{s+1:t}\boldsymbol{\xi}_s\right\|\right] \overset{(21)}{=} \frac{1}{B}\mathbb{E}\left[\left\|\sum_{s=1}^{t}\sum_{i=1}^{B}(1 - \beta_s)\beta_{s+1:t}\boldsymbol{\xi}_s^i\right\|\right] = \frac{1}{B}\mathbb{E}\left[\left\|\sum_{s=1}^{t}\sum_{i=1}^{B}\boldsymbol{v}_{(s-1)B+i}\right\|\right]
$$

$$
\leq \frac{2\sqrt{2}}{B}\mathbb{E}\left[\left(\sum_{s=1}^{t}\sum_{i=1}^{B}\left\|\boldsymbol{v}_{(s-1)B+i}\right\|^{\mathfrak{p}}\right)^{\frac{1}{\mathfrak{p}}}\right]
$$

$$
= \frac{2\sqrt{2}}{B}\mathbb{E}\left[\left(\sum_{s=1}^{t}\sum_{i=1}^{B}(1 - \beta_s)^{\mathfrak{p}}(\beta_{s+1:t})^{\mathfrak{p}}\left\|\boldsymbol{\xi}_s^i\right\|^{\mathfrak{p}}\right)^{\frac{1}{\mathfrak{p}}}\right]. \tag{26}
$$

Observe that

$$
\mathbb{E}\left[\left(\sum_{s=1}^{t}\sum_{i=1}^{B}(1 - \beta_s)^{\mathfrak{p}}(\beta_{s+1:t})^{\mathfrak{p}}\left\|\boldsymbol{\xi}_s^i\right\|^{\mathfrak{p}}\right)^{\frac{1}{\mathfrak{p}}} \mid \mathcal{F}_{t-1}\right]
$$

$$
\overset{(e)}{\leq} \left(\mathbb{E}\left[\sum_{s=1}^{t}\sum_{i=1}^{B}(1 - \beta_s)^{\mathfrak{p}}(\beta_{s+1:t})^{\mathfrak{p}}\left\|\boldsymbol{\xi}_s^i\right\|^{\mathfrak{p}} \mid \mathcal{F}_{t-1}\right]\right)^{\frac{1}{\mathfrak{p}}}
$$

$$
= \left(\mathbb{E}\left[\sum_{i=1}^{B}(1 - \beta_t)^{\mathfrak{p}}(\beta_{t+1:t})^{\mathfrak{p}}\left\|\boldsymbol{\xi}_t^i\right\|^{\mathfrak{p}} \mid \mathcal{F}_{t-1}\right] + \sum_{s=1}^{t-1}\sum_{i=1}^{B}(1 - \beta_s)^{\mathfrak{p}}(\beta_{s+1:t})^{\mathfrak{p}}\left\|\boldsymbol{\xi}_s^i\right\|^{\mathfrak{p}}\right)^{\frac{1}{\mathfrak{p}}}
$$

$$
\overset{\text{Assumption 2.4}}{\leq} \left(B(1 - \beta_t)^{\mathfrak{p}}(\beta_{t+1:t})^{\mathfrak{p}}(\sigma_0^{\mathfrak{p}} + \sigma_1^{\mathfrak{p}}\left\|\nabla F(\boldsymbol{x}_t)\right\|^{\mathfrak{p}}) + \sum_{s=1}^{t-1}\sum_{i=1}^{B}(1 - \beta_s)^{\mathfrak{p}}(\beta_{s+1:t})^{\mathfrak{p}}\left\|\boldsymbol{\xi}_s^i\right\|^{\mathfrak{p}}\right)^{\frac{1}{\mathfrak{p}}}
$$

$$
\leq \left(B(1 - \beta_t)^{\mathfrak{p}}(\beta_{t+1:t})^{\mathfrak{p}}\sigma_0^{\mathfrak{p}} + \sum_{s=1}^{t-1}\sum_{i=1}^{B}(1 - \beta_s)^{\mathfrak{p}}(\beta_{s+1:t})^{\mathfrak{p}}\left\|\boldsymbol{\xi}_s^i\right\|^{\mathfrak{p}}\right)^{\frac{1}{\mathfrak{p}}}
$$

$$
+ B^{\frac{1}{\mathfrak{p}}}(1 - \beta_t)\beta_{t+1:t}\sigma_1\left\|\nabla F(\boldsymbol{x}_t)\right\|, \tag{27}
$$

where $(e)$ is by Hölder's inequality. Taking expectations on both sides of (27) to get

$$
\mathbb{E}\left[\left(\sum_{s=1}^{t}\sum_{i=1}^{B}(1 - \beta_s)^{\mathfrak{p}}(\beta_{s+1:t})^{\mathfrak{p}}\left\|\boldsymbol{\xi}_s^i\right\|^{\mathfrak{p}}\right)^{\frac{1}{\mathfrak{p}}}\right]
$$

$$
\leq \mathbb{E}\left[\left(B(1 - \beta_t)^{\mathfrak{p}}(\beta_{t+1:t})^{\mathfrak{p}}\sigma_0^{\mathfrak{p}} + \sum_{s=1}^{t-1}\sum_{i=1}^{B}(1 - \beta_s)^{\mathfrak{p}}(\beta_{s+1:t})^{\mathfrak{p}}\left\|\boldsymbol{\xi}_s^i\right\|^{\mathfrak{p}}\right)^{\frac{1}{\mathfrak{p}}}\right]
$$

$$
+ B^{\frac{1}{\mathfrak{p}}}(1 - \beta_t)\beta_{t+1:t}\sigma_1\mathbb{E}\left[\left\|\nabla F(\boldsymbol{x}_t)\right\|\right].
$$

Recursively applying the above argument from $\mathcal{F}_{t-2}$ to $\mathcal{F}_0$, we can finally obtain

$$
\mathbb{E}\left[\left(\sum_{s=1}^{t}\sum_{i=1}^{B}(1 - \beta_s)^{\mathfrak{p}}(\beta_{s+1:t})^{\mathfrak{p}}\left\|\boldsymbol{\xi}_s^i\right\|^{\mathfrak{p}}\right)^{\frac{1}{\mathfrak{p}}}\right]
$$

$$
\leq B^{\frac{1}{\mathfrak{p}}}\left(\sum_{s=1}^{t}(1 - \beta_s)^{\mathfrak{p}}(\beta_{s+1:t})^{\mathfrak{p}}\right)^{\frac{1}{\mathfrak{p}}}\sigma_0 + B^{\frac{1}{\mathfrak{p}}}\sum_{s=1}^{t}(1 - \beta_s)\beta_{s+1:t}\sigma_1\mathbb{E}\left[\left\|\nabla F(\boldsymbol{x}_s)\right\|\right], \tag{28}
$$

which gives us

$$\mathbb{E}\left[\left\|\sum_{s=1}^{t}(1-\beta_s)\beta_{s+1:t}\boldsymbol{\xi}_s\right\|\right]$$

$$\stackrel{(26)}{\leq}\frac{2\sqrt{2}}{B}\mathbb{E}\left[\left(\sum_{s=1}^{t}\sum_{i=1}^{B}(1-\beta_s)^{\mathfrak{p}}(\beta_{s+1:t})^{\mathfrak{p}}\left\|\boldsymbol{\xi}_s^i\right\|^{\mathfrak{p}}\right)^{\frac{1}{\mathfrak{p}}}\right]$$

$$\stackrel{(28)}{\leq}\frac{2\sqrt{2}}{B^{\frac{\mathfrak{p}-1}{\mathfrak{p}}}}\left(\sum_{s=1}^{t}(1-\beta_s)^{\mathfrak{p}}(\beta_{s+1:t})^{\mathfrak{p}}\right)^{\frac{1}{\mathfrak{p}}}\sigma_0+\frac{2\sqrt{2}}{B^{\frac{\mathfrak{p}-1}{\mathfrak{p}}}}\sum_{s=1}^{t}(1-\beta_s)\beta_{s+1:t}\sigma_1\mathbb{E}\left[\|\nabla F(\boldsymbol{x}_s)\|\right].\quad(29)$$

Combining (23), (24), (25) and (29), we finally obtain for any $t\in[T]$,

$$\mathbb{E}\left[\|\boldsymbol{\epsilon}_t\|\right]\leq\frac{2\sqrt{2}}{B^{\frac{\mathfrak{p}-1}{\mathfrak{p}}}}\left[\beta_{1:t}\left(\sigma_0+\sigma_1\left\|\nabla F(\boldsymbol{x}_1)\right\|\right)+\left(\sum_{s=1}^{t}(1-\beta_s)^{\mathfrak{p}}(\beta_{s+1:t})^{\mathfrak{p}}\right)^{\frac{1}{\mathfrak{p}}}\sigma_0\right]$$

$$+\sum_{s=2}^{t}\beta_{s:t}\left(L_0+L_1\mathbb{E}\left[\|\nabla F(\boldsymbol{x}_{s-1})\|\right]\right)\eta_{s-1}$$

$$+\frac{2\sqrt{2}}{B^{\frac{\mathfrak{p}-1}{\mathfrak{p}}}}\sum_{s=1}^{t}(1-\beta_s)\beta_{s+1:t}\sigma_1\mathbb{E}\left[\|\nabla F(\boldsymbol{x}_s)\|\right].$$

$\square$

