# OpenReview forum: "Nonconvex Stochastic Optimization under Heavy-Tailed Noises: Optimal Convergence without Gradient Clipping"
_ICLR.cc/2025/Conference — ICLR 2025 Poster_

### Official Review · Reviewer_3wrr · 2024-11-01

**Soundness:** 3
**Presentation:** 3
**Contribution:** 3
**Rating:** 6
**Confidence:** 3

**Summary:**

This paper studies the batched normalized stochastic gradient descent with momentum (Batched NSGDM)
and proves that convergence under heavy-tailed noise, when only $p$-th moment is finite where $p$ is between $1$ and $2$.
Convergence analysis is also obtained when the tail-index $p$ is unknown. The assumptions are relatively mild.
One ingredient in the proof is a novel expected inequality for the vector-valued martingale difference sequence,
which might be of independent interest.

**Strengths:**

(1) Traditionally, under heavy-tailed noise, convergence analysis is carried out under clipping. This paper no longer requires clipping, but rather works with normalized gradient, and brings a new angle and contribution to this field.

(2) The paper is well written and the intuition behind the results and proofs is well explained.

(3) Some novel inequality is derived (Lemma 4.3.) as a by-product of the proof, which might be useful in other contexts.

**Weaknesses:**

(1) As the paper mentions in the conclusion and future work section, it would be interesting to investigate the adaptive gradient methods in the future because batched NSGDM is not commonly used in practice.

(2) The paper should make it more clear whether batched NSGDM is proposed as a novel algorithm in this paper, or it already exists in the literature (if that's the case, some references should be provided).  If batched NSGDM is not commonly used or is first proposed in this paper, even though this paper is a theory paper, numerical experiments will significantly strengthen this paper because it can show that batched NSGDM works well under heavy-tailed noise.

(3) Taking batch-size $B=1$ is quite restrictive. If the paper can relax this assumption, it would make the theory stronger. If not, at least, it should be made more clear whether this restriction is due to the limitations from the proof technique, or the limitations of the algorithm itself.

**Questions:**

(1) In the abstract, it seems the word batched is never mentioned although you focus on batched NSGDM in the rest of the paper.

(2) It might be better to define $T$ in the abstract.

(3) In Section 1.1., "We initial the study'' should be ''We initiate the study''.

(4) It is nice that you provided examples in the appendix regarding your Assumption 2.4. I think it would be nicer if you can provide some references and comparisons to the literature since you mentioned "Assumption 2.4. is a relaxation of the traditional heavy-tailed noises assumption''

(5) In Algorithm 1, how do you define it when $m_{t}=0$?

(6) In Lemma 4.1., Lemma 4.2., it seems that you are working with general $\beta_{t}$, $\eta_{t}$,  does that mean you can extend your Theorem 3.2, Theorem 3.6. to allow momentum parameter and learning rate to be time-dependent as well?

---

### Official Review · Reviewer_jQHz · 2024-11-02

**Soundness:** 2
**Presentation:** 3
**Contribution:** 3
**Rating:** 6
**Confidence:** 4

**Summary:**

By revisiting the existing Normalized Stochastic Gradient Descent with Momentum (NSGDM) algorithm authors provide the first convergence result under heavy-tailed noises but without gradient clipping. They also establish the first optimal convergence rate in the case where the tail index p is unknown in advance, which is arguably the common scenario in practice. The results are quite good from mathematical point of view (up to a further generalization for rates in terms of probability of large deviations), but in my opinion are mainly interesting from theoretical point of view, not practical (e.g. I can not find experiments). Moreover, when the tail index p is unknown in advance it seems the result not necessarily to be optimal.
BTW It seems that these papers https://arxiv.org/pdf/2409.14989, https://arxiv.org/pdf/2410.13849 could be added in to the literature survey.

**Strengths:**

I guess the paper is good from mathematical point of view. The results is accurate, original, well presented.

**Weaknesses:**

1) This is not the honest normalization procedure that is of independent interest. The motivation to replace clipping by something else is not very attractive for me. For example, under different symmetrical assumptions on noise you can think of different medians procedure rather that clipping, or sign SGD type procedures...
2) Many modern paper on heavy-tails SGD methods include analysis with high-probability deviations bounds, rather than in considered paper
3) Lack of numerical experiments. Weak motivation to consider convergence in terms of the norm of the gradient for real non-convex applications.

**Questions:**

Is it possible to generalize the result for convex problems? I can make rating significantly higher if you can add this case, but it seems to be more difficult one in my opinion (hope that you can dissuade me).

---

### Official Review · Reviewer_tQr8 · 2024-11-03

**Soundness:** 3
**Presentation:** 3
**Contribution:** 3
**Rating:** 8
**Confidence:** 4

**Summary:**

The paper addresses a problem of (generalized) smooth non-convex optimization with heavy-tailed noise in the stochastic gradients and provides new convergence guarantees for the Normalized SGD with Momentum (NSGDM). In particular, the authors assume $(L_0,L_1)$-smoothness, which is strictly more general than standard smoothness, of the objective function, its lower boundedness (classical assumption), and boundedness of the $p$-th moment of the gradient noise for some $p \in (1,2]$, i.e., $\mathbb{E}\|\| g(x) - \nabla F(x) \|\|^p \leq \sigma_0^p + \sigma_1^p \|\| \nabla F(x)\|\|^p$ for all $x \in \mathbb{R}^d$, where $F$ is the objective function. Then, the authors provide new results for (mini-batched) NSGDM - a method proposed by Cutcosky & Mehta (2020) - under these assumptions.

The first result establishes $O(T^{-\frac{p-1}{3p-2}})$ convergence rate for $\frac{1}{T}\sum_{t=1}^T\mathbb{E}\|\| \nabla F(x_t) \|\|$ in the case when $L_0, L_1, \sigma_0, \sigma_1, \Delta_1$, and, more importantly for the paper, $p$ are known. This result matches the known lower bound (when $L_1 = 0$ and $\sigma_1 = 0$) and the best-known rates for Clipped SGD. Moreover, the result extends the known ones to the case of $L_1 > 0$ and $\sigma_1 > 0$.

The second result establishes $O(T^{-\frac{p-1}{2p}})$ convergence rate but without any prior knowledge about all parameters except of $L_1$. This rate is worse than optimal but still matches it when $p = 2$ (since it is obtained for the parameters with the dependency on $T$ as in the case of $p=2$). This result demonstrates a noticeable robustness of NSGDM.

The authors also provide a refined formulation of the lower bound from Zhang et al. (2020), though the worst-case example and the proof are largely dependent on the previous results, i.e., on the proofs from Arjevani et al. (2023).

**Strengths:**

S1. The paper addresses an important question and makes an important contribution to the literature on stochastic optimization with heavy-tailed noise. In particular, I would like to highlight Theorem 3.6, which gives the first convergence result in the considered setting without the knowledge of $p$, and Lemma 4.3, which generalizes Lemma 7 from [1] and gives an alternative and quite simple proof to a similar result.

S2. The generality of the results is also noticeable. While most of the existing works on the stochastic optimization with the heavy-tailed noise focus on smooth problems, this paper considers $(L_0,L_1)$-smooth setup, which is known to better reflect the properties of the minimization problems arising in the training of neural networks. Moreover, bound on the $p$-th moment of the noise contains an additive term proportional to $\|\| \nabla F(x)\|\|^p$, while the previous works assume uniform boundedness of the $p$-th moment of the noise.

S3. The paper is, in general, well-written and easy to follow. I also did not find any mistakes in the proofs (though some parts could be further improved). Moreover, the authors provide intuitive explanations in the main text and the sketch of the proof, allowing to understand the key steps of the proof and also to see what the main technical novelty of the paper (which I believe is Lemma 4.3; other parts of the proof are very similar to the previous works such as [2, 3]; nevertheless, it does not undermine the contribution of the paper, in my opinion).



---
References

[1] Kornilov et al. Accelerated Zeroth-order Method for Non-Smooth Stochastic Convex Optimization Problem with Infinite Variance. NeurIPS 2023

[2] Cutckosky & Mehta. Momentum Improves Normalized SGD. ICML 2020

[3] Hübler et al. Parameter-agnostic optimization under relaxed smoothness. AISTATS 2024

**Weaknesses:**

W1. When $\sigma_1 > 0$, the result of Theorem D.1 requires the knowledge of $\|\| \nabla F(x_1) \|\|$. Moreover, the theorem is formulated in such a way that all problem-related parameters are required to be known precisely.

W2. When $\sigma_1 > 0$, the results hold only for $B > 1$.

W3. The paper lacks numerical experiments. In particular, it would be very interesting to see in some (even toy) experiments how NSGDM depends on the choice of $\beta$ and whether the rate $O(T^{-\frac{p-1}{2p}})$ is indeed tight *for NSGDM* when $p$ is unknown.

Nevertheless, I believe that the strengths outweigh the weaknesses.

**Questions:**

**Questions.**

Q1. Is it possible to prove a version of Theorem D.1 when only some upper bounds on problem-related parameters are known, i.e., upper bounds on $L_0, L_1, \sigma_0, \sigma_1, \Delta_1, \|\| \nabla F(x_1) \|\|$?

Q2. Is it possible to generalize the results of the paper to the case of $B=1$ even when $\sigma_1 > 0$? If no, could the authors provide a counter-example or an experiment illustrating that $B$ has to be larger than $1$ in this case?

Q3. Is the result for NSGDM without the prior knowledge of $p$ tight?

Q4. What is the difference between Theorem 3.3 and the result from [1]? As far as I see, the proofs are almost identical to the one from [2] (up to the difference in the choice of $q$).

Q5. Is it possible to generalize the results of this paper to the case when parameters are horizon-independent, i.e., independent of the total number of steps $T$, like in [3]?

Q6. Could the authors comment on the differences between Lemma 4.3 and Lemma 7 from [4] and add this discussion to the paper?

Q7. Line 809: why is the assumption with $\|\| x - y\|\|| \leq c/L_1$ is stronger than the one for $\|\| x-y \|\| \leq 1/L_1$? Aren't they both equivalent (up to the redefinition of $L_0$ and $L_1$) to the so-called symmetric $(L_0,L_1)$-smoothness from [5]?



**Comments.**

C1. I recommend adding the statements of the main results in the full generality to the main text, i.e., with $\sigma_1 > 0$. In my opinion, they are quite understandable for the broad audience, so they can be placed in the main text.

C2. Lines 104-106: the rate from Nguen et al. (2023) in the noiseless regime is $O(T^{-\frac{2p-1}{3p-2}})$ (for the average of the expected squared norms of the gradients), which is worse than $O(T^{-\frac{1}{2}})$.

C3. I believe that the discussion of the related work on $(L_0,L_1)$-smoothness should be extended. In particular, the authors should discuss how their results can be compared to the known results for NSGDM from [3].

C4. Line 334, "following the existing literature": the authors should provide the reference to the paper.

C5. Sketch of the proof of Lemma 4.2 on page 8: $m_0$ is undefined. I assume that $m_0 = g_1$.

C6. I suggest adding in (10) more details about the last step (that the authors used $\sum_{t}a_t b_t \leq (\sum_t a_t)(\sum_t b_t)$ for non-negative sequences).

C7. The last step in (13) is a bit technical and requires the reader to do a lot of calculations on their own. I suggest adding more details on how you estimate each term.

C8. Regarding the proof of Theorem 3.3: essentially, the proof is almost identical to the one from [2] up to the replacement of $\sigma$ with $\frac{\sigma_0^{\frac{p}{2(p-1)}}}{4\gamma \varepsilon^{\frac{p}{2(p-1)}-1}}$ because for the considered example all moments exist (though they are different). I also encourage the authors to mention explicitly that the construction in the lower bound depends on the target accuracy $\varepsilon$.




**Minor comments.**

M1. Line 16: noise, that --> noise, which

M2. Line 48: Simsekli et al. 2019  -- the same reference appear twice

M3. Line 108: should be "the results from Cutkosky & Mehta (2021); Liu et al. (2024) also depend"

M4. Line 256: Theorem 3.2 is not perfectly adaptive to $\sigma_0$ since it requires its knowledge. I suggest to rewrite this sentence.

M5. Line 324: help --> helps

M6. Line 325: smooth --> smoothness

M7. Title of Appendix C: 4.4 --> 4.3

M8. Line 855: AM-GM - better to replace abbreviation with a full version.

---
References

[1] Zhang et al. Why are Adaptive Methods Good for Attention Models? NeurIPS 2020.

[2] Arjevani et al. Lower bounds for non-convex stochastic optimization. Mathematical Programming 2023.

[3] Hübler et al. Parameter-agnostic optimization under relaxed smoothness. AISTATS 2024

[4] Kornilov et al. Accelerated Zeroth-order Method for Non-Smooth Stochastic Convex Optimization Problem with Infinite Variance. NeurIPS 2023

[5] Chen et al. Generalized-smooth nonconvex optimization is as efficient as smooth nonconvex optimization. ICML 2023

---

### Official Review · Reviewer_R9Yi · 2024-11-07

**Soundness:** 3
**Presentation:** 3
**Contribution:** 3
**Rating:** 8
**Confidence:** 3

**Summary:**

This paper addresses two questions:
- Is gradient clipping the only way to guarantee convergence under heavy-tailed noises? Can an alternate algorithm achieve the optimal convergence?
- Does there exist an algorithm that provably converges under heavy-tailed noises even if the tail index p is unknown?

**Strengths:**

- The paper is well-written. The results are well-explained.

**Weaknesses:**

- The paper is well-written, but, given the proof-heavy nature of the paper, it is difficult to follow a few things without looking at the Appendix. This is not necessarily a criticism of the paper, but a limitation of the format.

**Questions:**

- In line 336, how is $\mathbb E \langle \epsilon_t^b, \epsilon_t^u \rangle = 0$?
- In both Theorem 3.2 and 3.6, batch size $B=1$. Is there any case where a larger batch-size is needed for convergence?
- lines 300-304 - can you explain this point about needing $p$ when $\sigma_1 > 0$ a bit more?

---

### Meta-Review · Area_Chair_6CSf · 2024-12-21

**Metareview:**

This paper examines the convergence of Normalized Stochastic Gradient Descent with Momentum (NSGDM) for heavy-tailed noise, where the $p$-th order moment is bounded for $p \in (1,2]$. The authors present the first results demonstrating that NSGDM converges at an optimal rate without requiring gradient clipping, as used in prior works. Additionally, they provide the first convergence results for cases where the index $p$ is unknown. All reviewers agree on the technical soundness and novelty of the paper. Some raised concerns about the lack of empirical validation of the algorithm, insufficient comparisons with related works addressing heavy-tailed SGD-type algorithms, and questions about the tightness of the results. During the rebuttal phase, the authors successfully addressed most of these concerns. Given the paper’s theoretical contributions and novelty, I support its publication. The authors should include the promised discussions and revisions in the final version.

**Additional Comments On Reviewer Discussion:**

Some reviewers raised concerns about the lack of empirical validation of the algorithm, insufficient comparisons with related works that employ similar techniques and bounds for heavy-tailed SGD-type algorithms, and the tightness of the results. During the rebuttal phase, the authors successfully addressed most of these concerns. However, the questions of empirical validation and the tightness of the results in the unknown index setting remain open and are left for future study.

---

### Decision · Program_Chairs · 2025-01-22

Accept (Poster)